# Research on Photon-Integrated Interferometric Remote Sensing Image Reconstruction Based on Compressed Sensing

**Jiawei Yong** [1], **Kexin Li** [2], **Zhejun Feng** [1], **Zengyan Wu** [1], **Shubing Ye** [1], **Baoming Song** [1], **Runxi Wei** [1] **and Changqing Cao** [1,*]

[1] School of Optoelectronic Engineering, Xidian University, 2 South Taibai Road, Xi'an 710071, China; 20051212225@stu.xidian.edu.cn (J.Y.); zhjfeng@mail.xidian.edu.cn (Z.F.); zywu_21@stu.xidian.edu.cn (Z.W.); sbye@stu.xidian.edu.cn (S.Y.); 21051212268@stu.xidian.edu.cn (B.S.); 21051212248@stu.xidian.edu.cn (R.W.)

[2] China Siwei Surveying and Mapping Technology Co., Ltd., 5 Fengxian East Road, Haidian District, Beijing 100094, China; likexin@chinasiwei.com

[*] Correspondence: chqcao@mail.xidian.edu.cn

**Abstract:** Achieving high-resolution remote sensing images is an important goal in the field of space exploration. However, the quality of remote sensing images is low after the use of traditional compressed sensing with the orthogonal matching pursuit (OMP) algorithm. This involves the reconstruction of the sparse signals collected by photon-integrated interferometric imaging detectors, which limits the development of detection and imaging technology for photon-integrated interferometric remote sensing. We improved the OMP algorithm and proposed a threshold limited-generalized orthogonal matching pursuit (TL-GOMP) algorithm. In the comparison simulation involving the TL-GOMP and OMP algorithms of the same series, the peak signal-to-noise ratio value ($P_{SNR}$) of the reconstructed image increased by 18.02%, while the mean square error ($M_{SE}$) decreased the most by 53.62%. The TL-GOMP algorithm can achieve high-quality image reconstruction and has great application potential in photonic integrated interferometric remote sensing detection and imaging.

**Keywords:** remote sensing image; compressed sensing; image reconstruction; photon-integrated technology; detection image

## 1. Introduction

With the increasingly mature manufacturing process of photonic integrated devices and interference detection technology, the segmented planar imaging detector for electro-optical reconnaissance (SPIDER), which has photonic integrated interference imaging as its core technology, has attracted a lot of attention from researchers in the field of astronomical observation or remote sensing detection. It has been used to replace traditional optical telescopes with large volume, weight, and energy consumption [1] in the detection of targets. For example, the Hubble Telescope is 13.3 m long and weighs 27,000 pounds [2].

Interferometry is an important technology used in photonic integrated interferometric imaging systems. It uses electromagnetic wave superposition to extract the wave source information and provides technical support for the reconstruction of high-resolution images. Optical interferometry unifies the light from many lens pairs on a photonic integrated chip (PIC) and then reconstructs the remote sensing image from the optical signal obtained by interferometry. Optical interferometer arrays are the preferred instruments for high-resolution imaging. Such interferometer arrays include the CHARA array [3,4], larger telescope interferometer [5], and navy precision optical interferometer [6]. These systems use far-field spatial coherence measurements to form intensity images of light source targets [7]. In our previous publication [8,9], we discussed the definition of a small-scale interferometric imager, which we called a planar photoelectric detection imaging detector (SPIDER). The SPIDER imager [8] comprises one-dimensional interferometric arms arranged along the azimuth angles in multiple directions. Each interference arm has

the same design structure. Any two lenses on the interference arm form the interference baseline; the collected optical signals are coupled in the PIC and interfered in the multimode interferometer (MMIs), while the fringe data are read by the two-dimensional detector array. Because the interference arms in the PIC are distributed along the azimuth angle $[0, 2\pi]$, and since interference baselines of any length on the interference arms correspond to the spatial frequency information in the two-dimensional Fourier Transform domain, the PIC can obtain optical frequency information through sparse sampling in all directions. We can use the compressing sensing (CS) theory algorithm to reconstruct sparse optical signal data in order to obtain the content information of detection targets. The CS theory can be applied in the field of photonic integrated interference imaging to meet our needs in life, production, and scientific exploration.

In recent years, CS has attracted increasing attention in signal processing. Donoho et al. proposed this theory in 2006. The traditional Nyquist sampling theorem [10] requires that the sampling frequency of the signal be greater than or equal to twice the signal frequency. The proposed compressed sensing theory overcomes the limitations of traditional sampling theorems. If the collected signals are sufficiently sparse, the original signals can be reconstructed by projection onto random vectors. More specifically, the original signals could be reconstructed at low speeds. Therefore, this innovative theory of improving sampling efficiency has been of great interest in the fields of digital signal processing [11], optical imaging [12], medical imaging [13], radio communication [14], radar imaging [15], and pattern recognition [16]. The research conducted on compressed sensing comprises three main areas: (1) the sparse representation of original signals, with commonly used sparse transform methods such as the Fourier Transform (FT) [17], Discrete Cosine Transform (DCT) [18], and Wavelet Transform (DWT) [19]; (2) the design of the measurement matrix, including the random measurement matrix [20,21] and deterministic measurement matrix [22,23]; (3) reconstruction algorithms, such as the basis pursuit (BP) algorithm [24,25], matching pursuit (MP) algorithm [26], and orthogonal matching pursuit algorithm [27–30].

The compressed sensing OMP algorithm is one of the most representative greedy algorithms; it is simple, stable, has low computational complexity, and has been widely studied by researchers. In contrast, the traditional OMP algorithm produces Gaussian noise when reconstructing an image, which significantly affects the quality of the reconstructed image. Consequently, the traditional OMP algorithm has continuously been improved over time, and enhanced algorithms such as stagewise orthogonal matching pursuit (STOMP), generalized orthogonal matching pursuit (GOMP), and stagewise weak orthogonal matching pursuit (SWOMP) have been generated to improve the quality of the reconstructed image. To further solve the above-mentioned problems, we improved the threshold limited-generalized orthogonal matching tracing algorithm using the traditional OMP algorithm.

The main contributions of this paper can be summarized as follows:

(1) We improved the traditional OMP algorithm and proposed the TL-GOMP algorithm, which was used to reconstruct the sparse spatial frequency information collected by the PIC and recover the content information of the detected target. In the simulation, we compared the TL-GOMP algorithm with the other improved OMP image reconstruction algorithm from the same series and the non-OMP image reconstruction algorithm, and subsequently verified its superiority in image reconstruction.
(2) Simultaneously, we used this algorithm to reconstruct and simulate the sparse signals collected by photonic integrated chips at different distances. The simulation results showed that the TL-GOMP algorithm can be applied in the field of photon-integrated interferometric remote sensing detection and imaging to recover the content information of unknown targets.

## 2. Related Work

Image reconstruction is based on sparse original signals from the target or image, and the content and feature information of the target or image are restored and reproduced by

designing reconstruction algorithms. At present, the compressed sensing reconstruction algorithm has become the mainstream in the field of image reconstruction, mainly because the image signal has two characteristics: high dimension and can be sparse. The research on compressed sensing theory mainly includes three aspects: sparse signal representation, measurement matrix design, and reconstruction algorithm design.

### 2.1. Sparse Signal Representation

The sparse representation of signals is an important premise and foundation of compressed sensing theory. When a signal can become approximately sparse under the action of a change domain, it is said to have sparsity or compressibility, which can achieve the purpose of reducing signal storage space and effectively compressed sampling. If the length of a signal is N, and the number of non-zero value elements is no more than k after representation by the sparse basis matrix, we can define it as a k-sparse signal. The sparsity k of the sparse signal directly affects the accuracy of the reconstructed signal; that is, the higher the sparsity, the higher the accuracy of the reconstructed signal. Based on the above reasons, the reasonable selection of the sparse basis matrix is very important. The commonly used transform bases are as follows: Fourier Transform basis [17], Discrete Cosine Transform basis [18], Discrete Wavelet Transform basis [19], Contourlet Transform basis [31], and the K-singular value decomposition method based on matrix decomposition [32].

### 2.2. Design of Measurement Matrix

In compressed sensing theory, the measurement matrix has the function of sampling the original signal, and its selection is very important. It can project the signal from a high-dimensional space to a low-dimensional space to obtain the corresponding measurement value. In order to obtain an accurate sparse representation through measurement values, an uncorrelated relationship between the observed matrix and the sparse basis matrix was required to satisfy the Restricted Isometry Property (RIP), which guaranteed that the original space and the sparse space could be mapped one-to-one. At the same time, the matrix formed by arbitrarily extracting the number of column vectors that was equal to the number of observed values is non-singular. Commonly used measurement matrices are as follows: Gaussian random matrix [33], measurement matrix constructed based on equilibrium Gold sequence [34], partial Fourier matrix [35], and partial Hadamard matrix [36]. Wang Xia proposed a deterministic random sequence measurement matrix [37] and verified its effectiveness through experiments.

### 2.3. Design of Reconstruction Algorithm

In recent years, remarkable achievements have been made in the research on compressed sensing reconstruction algorithms, which can be divided into the traditional iterative compressed sensing reconstruction algorithm and the deep compressed sensing network-based reconstruction algorithm.

#### 2.3.1. Traditional Iterative Compressed Sensing Reconstruction Algorithm

The purpose of compressed sensing is to find the sparsest original signal to meet the demands of measurement, which can be understood as the inverse problem of minimizing the norm of l0. Specific methods for achieving this are as follows: (1) convex relaxation method, which converts the minimum l0-norm problem into the minimum l1 norm problem under certain conditions, that is, the non-convex problem is converted into a convex problem such as the basis pursuit algorithm [38] and the Gradient Projection for Sparse Reconstruction (GPSR) [39]; (2) greedy matching tracking algorithms such as the matching pursuit algorithm [26] and the orthogonal matching pursuit algorithm [27,28]; (3) non-convex optimization methods, including the Bayesian Compressed Sensing algorithm (BCS) [40]; and (4) model-based optimization algorithms, the first three of which are based on the sparsity of original signals; these may not be valid for ordinary signals, such as the improved Total Variation-based algorithm (TV) [41].

2.3.2. Reconstruction Algorithm Based on Deep Compressed Sensing Network

As the use of deep learning in various research fields has increased, it has gradually been introduced into the research on compressed sensing image reconstruction algorithms. Ali Mousavi et al. proposed a Stacked Denoising Autoencode (SDA) algorithm, which mainly realized the end-to-end mapping between measured values and reconstructed images and adopted an unsupervised learning method. Kulkarni et al. proposed Recon-Net [42], a non-iterative framework based on convolutional neural networks, and applied convolutional neural networks to compressed sensing reconstruction for the first time. The network structure consisted of a fully connected layer and six convolutional layers. Yao and Dai et al. combined the idea of residual learning with ReconNet and proposed a Deep Residual Reconstruction Network (DR2-Net) for compressed image perception reconstruction [43]; the network was cascaded. Kulkarni K, Lohit S et al. [44] further deepened the network structure of ReconNet and used the network structure of a full connection layer to replace the original Gaussian matrix in order to realize the image sampling. This kind of network is called adaptive sampling ReconNet. Xuemei Xie et al. [45] made some improvements to the sampling process of compressed sensing and also used full connection and deconvolution methods to optimize the compressed sensing network. Nie and Fu et al. not only used the convolutional neural network for image reconstruction, but also added image denoising into the network. The ResConv network they proposed [46] has these two characteristics. The CSnet network proposed by Shi et al. [47] redesigned the sampling process, which, as with the previous algorithms, does not only realize image reconstruction, but also puts forward a novel sampling mechanism to match the reconstructed network.

## 3. Methods

The theoretical framework of compressed sensing consists of three main aspects: the sparse representation of the original signal vector $\vec{x}$; the measurement matrix designed to change the high-dimensional original signal into a low-dimensional measurement vector $\vec{y}$; and the algorithm designed to obtain the approximate sparse representation $\vec{\theta}$ in order to recover the original signal.

### 3.1. The Reconstruction Principle of the OMP Algorithm Based on Compressed Sensing

Figure 1 shows a schematic for solving sparse representations in compressed sensing. Here, we consider the compressed sensing theory as a linear model:

$$\left[\vec{y_1}, \vec{y_2}, \cdots\cdots, \vec{y_n}\right] = B_{m \times n} \times \left[\vec{x_1}, \vec{x_2}, \cdots\cdots, \vec{x_n}\right] \tag{1}$$

where $\vec{y} \in R^m$ and $\vec{x} \in R^n$ represent the column vectors in the observation data and unknown image, respectively, and the measurement matrix $B \in R^{m \times n}$ arranged in the order of the column vectors is known. We chose the unknown image $D \in R^{n \times n}$, which can be represented by $D = \left[\vec{x_1}, \vec{x_2}, \cdots\cdots, \vec{x_n}\right]$.

Because the original signal $\vec{x}$ is not absolutely sparse, to transform it into a compressible signal, a sparse basis matrix $\Psi \in R^{n \times n}$ is adopted, which transforms the original signal into a sparse domain and forms a sparse representation vector $\vec{\theta} \in R^{n \times 1}$. The number of non-zero values in the sparse representation vector $\vec{\theta} = [k_1, k_2 \cdots\cdots k_{n-1}, 0, 0]$ is $k \ll n$, and thus the vector $\vec{\theta}$ is called the *k*-sparse representation. The measured data of the target can be written as follows:

$$\begin{aligned}
\left[\vec{y_1}, \vec{y_2}, \cdots\cdots, \vec{y_n}\right] &= B \times \Psi \times \left[\vec{\theta_1}, \vec{\theta_2}, \cdots\cdots, \vec{\theta_n}\right] \\
&= \begin{pmatrix} b_{11} & \cdots & b_{1n} \\ \vdots & \ddots & \vdots \\ b_{m1} & \cdots & b_{mn} \end{pmatrix} \times \begin{pmatrix} \psi_{11} & \cdots & \psi_{1n} \\ \vdots & \ddots & \vdots \\ \psi_{m1} & \cdots & \psi_{mn} \end{pmatrix} \times \left[\vec{\theta_1}, \vec{\theta_2}, \cdots\cdots, \vec{\theta_n}\right]
\end{aligned} \tag{2}$$

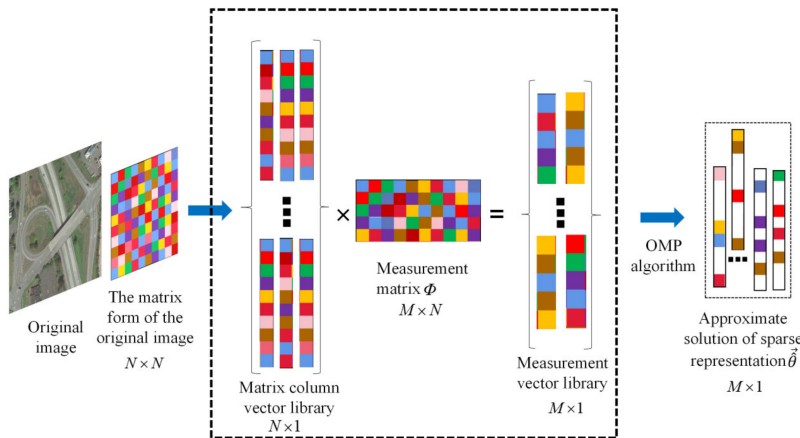

**Figure 1.** Schematic of the sparse representation of compressed sensing.

Here, we define the sensor matrix $A$, whose function is to establish the linear relationship between the sparse representation $\vec{\theta}$ and the measured value $\vec{y}$.

The measurement data of the target can then be expressed as

$$\left[\vec{y_1}, \vec{y_2}, \cdots\cdots, \vec{y_n}\right] = A \times \left[\vec{\theta_1}, \vec{\theta_2}, \cdots\cdots, \vec{\theta_n}\right] \tag{3}$$

Here, we use the most commonly used OMP algorithm [30,31] to illustrate the approximate solution $\vec{\theta}$ of the sparse representation. A column vector $\vec{y} \in \left[\vec{y_1}, \vec{y_2}, \cdots\cdots, \vec{y_n}\right]$ of the measurement data is selected, and $\vec{r}^{(k)}$ is used to represent the residual value after the $k_{TH}$ iteration. The initial value of the residual is set as $\vec{r}^{(k)} = \vec{y}^{(0)}$. $\Lambda_k$ represents the matrix used to store the column vector $\vec{a}_k$ of the sensor matrix after the $k_{TH}$ iteration. The initial value of the matrix is represented by $\Lambda_0$. The sensor matrix is defined as follows:

$$A = \begin{pmatrix} a_{11} & \cdots & a_{1n} \\ \vdots & \ddots & \vdots \\ a_{m1} & \cdots & a_{mn} \end{pmatrix}_{m \times n} = \left[\vec{a}_1, \vec{a}_2 \cdots\cdots \vec{a}_n\right] \tag{4}$$

After multiplying the transposed form $\left[\vec{a}_1, \vec{a}_2 \cdots\cdots \vec{a}_n\right]^T$ of the sensor matrix $A$ with the initial residual value $\vec{r}^{(0)}$, $\vec{b}$ can be expressed as

$$\vec{b} = \begin{pmatrix} a_{11} & \cdots & a_{1m} \\ \vdots & \ddots & \vdots \\ a_{n1} & \cdots & a_{nm} \end{pmatrix}_{n \times m} \times \vec{y} = \begin{pmatrix} a_{11} & \cdots & a_{1m} \\ \vdots & \ddots & \vdots \\ a_{n1} & \cdots & a_{nm} \end{pmatrix}_{n \times m} \times \begin{bmatrix} y_1 \\ \vdots \\ y_m \end{bmatrix}_{m \times 1} = \begin{bmatrix} b_1 \\ \vdots \\ b_m \end{bmatrix}_{m \times 1} \tag{5}$$

Here, each element in the vector $\vec{b}^T = [b_1, b_2 \cdots\cdots b_m]$ represents the inner product of each row vector in $\left[\vec{a}_1, \vec{a}_2 \cdots\cdots \vec{a}_n\right]^T$ with $\vec{r}^{(0)}$, that is, $b_i = \vec{a}_j^T \times \vec{r}^{(0)}$ ($i = 1, 2 \cdots\cdots m$; $j = 1, 2 \cdots\cdots n$). The corresponding column vector $\vec{a}_j$ in the sensor matrix is selected according to the maximum inner product value $b_i$, and $\vec{a}_j$ is stored in the $\Lambda$ matrix. The least-squares method is used to obtain the minimum residual value $\vec{c}^{(k)} = \left(\vec{a}_j^T \times \vec{a}_j\right)^{-1} \times \vec{a}_j^T \times \vec{y}^{(k-1)}$. The residual value $\vec{r}^{(k)}$ after the $k_{TH}$ iteration is

$$\vec{r}^{(k)} = \vec{y}^{(k-1)} - \vec{a}_j \times \vec{c}^{(k)} \tag{6}$$

where $\vec{a}_j^{(k)}$ represents the column vector selected from the sensor matrix during the $k_{TH}$ iteration. Finally, after $k$ iterations, we can obtain the $k$-sparse representation (approximate solution $\vec{\theta}$), which comprises $k$ non-zero values such as $c^{(1)}, c^{(2)} \cdots\cdots c^{(k)}$. This is an optimization problem for the smallest norm of $l_1$, which can be mathematically expressed as follows:

$$\min_{\vec{\theta}} \left\| \vec{\theta} \right\|_{l_1} s.t. \vec{y} = B\psi\vec{\theta} \tag{7}$$

Algorithm 1 presents the execution steps of the OMP algorithm. As shown in Figure 2, we multiply the approximate solution $\vec{\theta}$ by the sparse basis matrix $\psi$; then, the original signal recovered is $\vec{\hat{x}} = \Psi \times \vec{\hat{\theta}}$. The final reconstructed image is obtained as follows:

$$\left[\vec{\hat{x}}_1, \vec{\hat{x}}_2, \cdots\cdots, \vec{\hat{x}}_n\right] = \Psi \times \left[\vec{\hat{\theta}}_1, \vec{\hat{\theta}}_2, \cdots\cdots, \vec{\hat{\theta}}_n\right] \tag{8}$$

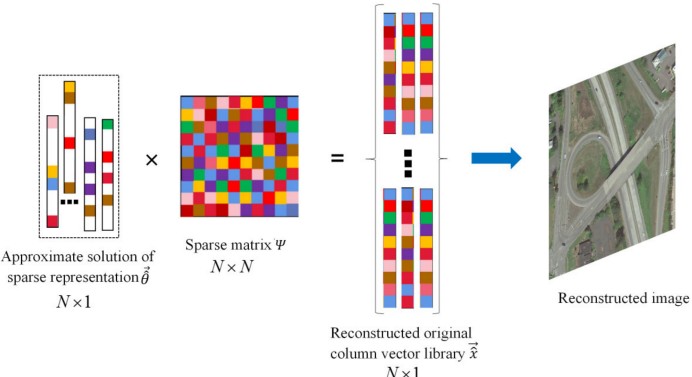

Approximate solution of sparse representation $\vec{\hat{\theta}}$
$N \times 1$

Sparse matrix $\Psi$
$N \times N$

Reconstructed original column vector library $\vec{\hat{x}}$
$N \times 1$

Reconstructed image

**Figure 2.** Schematic of original image reconstruction using sparse representation $\vec{\hat{\theta}}$.

---

**Algorithm 1**: Orthogonal Matching Pursuit

---

**Input**: Sensor matrix $B$, Sparseness $k$

**Output**: Sparse representation $\vec{\theta}$

**Initialize:** Residual $\vec{r}_0 = \vec{y}$, Index set $\Lambda_0 = \varnothing$, $t = 1$

Loop performs the following five steps:

(1) Find out : $q : q_t = \operatorname{argmax}_{j=1\ldots\ldots N} \left| \left\langle \vec{r}_{t-1}, \alpha_j \right\rangle \right|$;

(2) Update the index set: $\Lambda_t = \Lambda_{t-1} \cup \{q_t\}$; Reconstruction of atomic collection: $B_t = [B_{t-1}, \alpha_q]$;

(3) Least-squares method: $\vec{\theta}_t = \operatorname{argmin} \left\| \vec{y} - B_t \vec{x} \right\|_2$;

(4) Update the residual: $\vec{r}_t = \vec{y} - B_t \vec{\theta}_t$, $t = t + 1$;

(5) Judgment: **If** $t > k$, stop the iteration, or go to step (1).

---

### 3.2. The Reconstruction Principle of the TL-GOMP Algorithm Based on Compressed Sensing

In this section, we introduce an improved TL-GOMP algorithm based on the traditional OMP algorithm. We selected unknown images $D \in R^{N \times N}$. To illustrate the principle of the improved TL-GOMP algorithm, we took the unknown target $D \in R^{N \times N}$ and converted it into the form of a column vector $D = \left[\vec{x}_1, \vec{x}_2 \cdots\cdots \vec{x}_n\right]$, as expressed by the equation below:

$$\left[\vec{y}_1, \vec{y}_2 \cdots\cdots \vec{y}_n\right] = \begin{pmatrix} b_{11} & \cdots & b_{1n} \\ \vdots & \ddots & \vdots \\ b_{m1} & \cdots & b_{mn} \end{pmatrix}_{m \times n} \times \left[\vec{x}_1, \vec{x}_2 \cdots\cdots \vec{x}_n\right] \tag{9}$$

Here, $\Psi = \begin{pmatrix} b_{11} & \cdots & b_{1n} \\ \vdots & \ddots & \vdots \\ b_{m1} & \cdots & b_{mn} \end{pmatrix}_{m \times n}$ is the measurement matrix. Assuming that the residual value after $k$ iterations is $\vec{r}^{(k)}$, we arbitrarily extracted a column $\vec{y} \in \left[ \vec{y}_1, \vec{y}_2 \cdots \cdots \vec{y}_n \right]$ and assigned it to the initial value $\vec{r}^{(0)} = \vec{y}$ of the residual value. By using the transpose form $\left[ \vec{a}_1, \vec{a}_2 \cdots \cdots \vec{a}_n \right]^T$ of the sensor matrix $B = \begin{pmatrix} a_{11} & \cdots & a_{1n} \\ \vdots & \ddots & \vdots \\ a_{m1} & \cdots & a_{mn} \end{pmatrix}_{m \times n} = \left[ \vec{a}_1, \vec{a}_2 \cdots \cdots \vec{a}_n \right],$

and multiplying the residual value $\vec{r}^{(k-1)} \in R^{m \times 1}$, we could obtain vector $\vec{d} \in R^{n \times 1}$ as follows:

$$\vec{d} = \begin{bmatrix} d_1 \\ \vdots \\ d_n \end{bmatrix} = \begin{pmatrix} a_{11} & \cdots & a_{1m} \\ \vdots & \ddots & \vdots \\ a_{n1} & \cdots & a_{nm} \end{pmatrix} \times \vec{r}^{(k-1)} \tag{10}$$

Here, we define one parameter $q_s = \left\| \vec{r}^{(k-1)} \right\| / \sqrt{M}$ and the other parameter $m_s = 1$; the parameter $m_s$ can be understood as a variable that controls or adjusts the threshold, which is a range value. The principle of its selection is to constantly change the threshold value and form the corresponding column vector of the first $S$ inner product values in the sensor matrix into a matrix, with the purpose of solving the optimal $S$ least-squares solutions to form a sparse representation. After $k$ cycles, the sparsity of the sparse representation is $kS$. Parameter $M$ represents the number of rows of perception matrix and measurement matrix in compressed sensing theory, or the number of measurements of observation matrix. The threshold $Th$ is then denoted as

$$Th = m_s q_s = \left\| \vec{r}^{(k-1)} \right\| / \sqrt{M} \tag{11}$$

Subsequently, we took the absolute value of each of the elements in the vector $\vec{d}$ and placed them in descending order to obtain the vector $\vec{d}^T = [d_{11}, d_{22} \cdots \cdots d_{nn}]$. We stored the sequence numbers of the elements satisfying the inequality relation in Equation (12).

$$\vec{d}^T = [d_{11}, d_{22} \cdots \cdots d_{nn}] \geq \left\| \vec{r}^{(k-1)} \right\| / \sqrt{M} \tag{12}$$

The algorithm cycles $k$ times in total, where $k$ refers to the number of non-zero-valued elements in the sparse representation. Each cycle will store the maximum number of elements $S$ that satisfy the threshold conditions. After $k$ cycles, there is a $kS$ value. Thereafter, the column vector of the sensor matrix corresponding to the value of $kS$ is stored in the matrix $A_t$, where $A_t \in R^{M \times kS}$. We then use the least-squares method to obtain the approximate solution $\vec{\hat{\theta}}$ for the sparse representation, as expressed by the equation below:

$$\vec{\hat{\theta}} = \left( A_t^T \times A_t \right)^{-1} \times A_t^T \times \vec{r}^{(k-1)} \tag{13}$$

After each iteration, the updated residual value $\vec{r}^{(k)}$ is expressed as:

$$\vec{r}^{(k)} = \vec{r}^{(k-1)} - A_t \times \vec{\hat{\theta}} \tag{14}$$

Finally, the reconstructed image $\hat{D} = \left[ \vec{\hat{x}}_1, \vec{\hat{x}}_2 \cdots \cdots \vec{\hat{x}}_n \right]$ is obtained as follows:

$$\hat{D} = \left[ \vec{\hat{x}}_1, \vec{\hat{x}}_2 \cdots \cdots \vec{\hat{x}}_n \right] = \begin{pmatrix} \psi_{11} & \cdots & \psi_{1n} \\ \vdots & \ddots & \vdots \\ \psi_{m1} & \cdots & \psi_{mn} \end{pmatrix} \times \left[ \vec{\hat{\theta}}_1, \vec{\hat{\theta}}_2 \cdots \cdots \vec{\hat{\theta}}_n \right] \tag{15}$$

As listed in Algorithm 2, the core function of the algorithm is to effectively select the maximum number $S$ using the additional threshold value. After the algorithm iterates $k$ times, the sparse representation vector has $kS$ sparsity. The threshold value used was $Th = m_s q_s$. In the subsequent simulations, we selected $m_s = 1$ and $q_s = \left\| \vec{r}^{(k-1)} \right\| / \sqrt{M}$.

---

**Algorithm 2:** Threshold Limited–Generalized Orthogonal Matching Pursuit

---

**Input:** Sensor matrix $B$, Sparseness $k$

**Output:** Sparse representation $\vec{\theta}$, Residual $\vec{r}_k$

**Initialize:** Residual $\vec{r}_0 = \vec{y}$, Index set $\Lambda_0 = \varnothing$, $A_0 = \varnothing$, $t = 1$

Loop performs the following five steps:

(1) Find out $q : q_t = \text{argmax}_{j=1\cdots\cdots N} \left| \left\langle \vec{r}_{t-1}, \alpha_j \right\rangle \right|$, selecting the maximum number $S$ of values that are greater than the threshold value $Th = m_s q_s$;

(2) Update the index set : $\Lambda_t = \Lambda_{t-1} \cup \{q_t\}$; Reconstruction of atomic collection $B_t = [B_{t-1}, \alpha_q]$;

(3) Least-squares method: $\vec{\theta}_t = \text{argmin} \left\| \vec{y} - B_t \vec{x} \right\|_2$;

(4) Update the residual : $\vec{r}_t = \vec{y} - B_t \vec{\theta}_t$, $t = t + 1$;

(5) Judgment: **If** $t > k$, stop the iteration, or go to step (1).

---

The advantage of this algorithm is that the inner product values meeting the threshold conditions can be quickly screened out in time by setting the limiting threshold value $Th = m_s q_s$, and corresponding column vectors can be directly found in the sensor matrix according to the serial number of the first $S$ inner product values. These inner product values are represented by logical value 1 in the code, while other inner product values are represented by logical value 0. On the other hand, by setting the threshold coefficient $m_s$ to adjust the limiting threshold, we constantly combine the serial numbers of the first $S$ inner product values into the corresponding column vectors in the sensor matrix to form a matrix, aiming at solving the optimal $S$ least-squares solutions with good universality and flexibility. After $k$ iterations, we can reconstruct the image information of the target through sparse representation $\vec{\theta}$ with a sparsity of $kS$.

## 4. Experiments

To demonstrate the performance of the TL-GOMP algorithm (refer to Algorithm 2) in reconstructing the target images, we present some simulation results in this section. First, we used the TL-GOMP algorithm and the same series of OMP, STOMP, and GOMP algorithms to conduct a comparative simulation of the target test images, as shown in Figure 3. In this case, the simulation results show that the image quality reconstructed using the TL-GOMP algorithm is better than that reconstructed using the same series of OMP algorithms. Subsequently, in order to rigorously prove the advantages of the TL-GOMP algorithm, we selected algorithms other than the OMP series to conduct a comparative simulation of the targets shown in Figure 3, and the simulation results once again showed that the image reconstructed by the TL-GOMP algorithm was better than that reconstructed by the other algorithms. We then applied the TL-GOMP algorithm in the field of photonic integrated interference image reconstruction and used this algorithm to reconstruct sparse spatial frequency information collected by the PIC at different distances. The simulation image results show that this algorithm can reconstruct the content information of the detected target

well. Finally, we explored the measurement number $M$ and sparsity $k$ in the TL-GOMP algorithm and their influence on the quality of the reconstructed images.

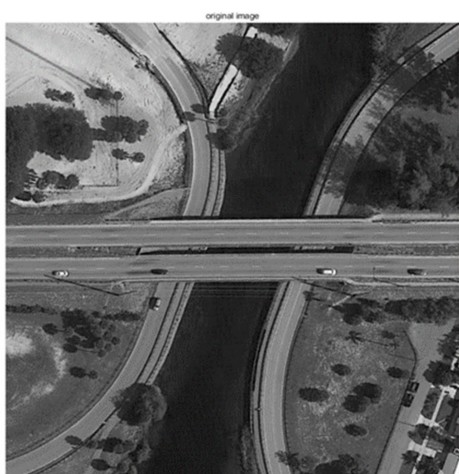

**Figure 3.** Original image.

In all the experiments below, we used the Gaussian random matrix as the measurement matrix, which is established by the randn function in the code, and the values of each element in this matrix satisfy the standard normal distribution. Meanwhile, we used the discrete cosine transform matrix as the sparse matrix, whose function is the sparse representation or compression of the original signal. In the experiments, the measurement matrix was updated with the operation of the code every time, which reflects the randomness. Therefore, we conducted several simulation experiments in each research part and verified the reliability of the conclusion through the data results.

### 4.1. Comparison of Simulation Results of the TL-GOMP and OMP Series Algorithms

For this section, we selected test images with pixel values of $350 \times 350$, $500 \times 500$, $650 \times 650$, and $800 \times 800$ as the target scenes; four image reconstruction algorithms (OMP, STOMP, GOMP, and TL-GOMP) to perform image reconstruction simulation; and used peak signal-to-noise ratio and mean square error to evaluate the image quality. Figure 4a–d show the simulation results of the $800 \times 800$ image reconstruction. From an intuitive point of view, the improved TL-GOMP tracing algorithm can be used to further improve the quality of the reconstructed images. Table 1 presents the quality evaluation data and the code runtime for the $350 \times 350$ reconstructed images. From the perspective of quantitative data, we can also see that the $P_{SNR}$ values of the images obtained by the TL-GOMP algorithm increased by 15.82% (compared with the results of the GOMP algorithm), 14.60% (compared with the STOMP algorithm), and 16.63% (compared with the OMP algorithm). The $M_{SE}$ values of the images decreased by 48.64%, 46.30%, and 50.12%, respectively, and the code running time was relatively fast. Therefore, from the above simulation data, we conclude that the TL-GOMP algorithm based on compressed sensing can rely on sparse data collected by the PIC to restore the content information of the detected target.

**Table 1.** $P_{SNR}$, $M_{SE}$, and time of the $350 \times 350$ image reconstructed by OMP series algorithms.

|  | OMP | STOMP | GOMP | TL-GOMP |
| --- | --- | --- | --- | --- |
| $P_{SNR}$ (dB) | 18.1671 | 18.4884 | 18.2944 | 21.1882 |
| $M_{SE}$ | 991.6670 | 920.9654 | 963.0243 | 494.6024 |
| Running time (s) | 3.3577 | 1.7663 | 2.7278 | 2.5031 |

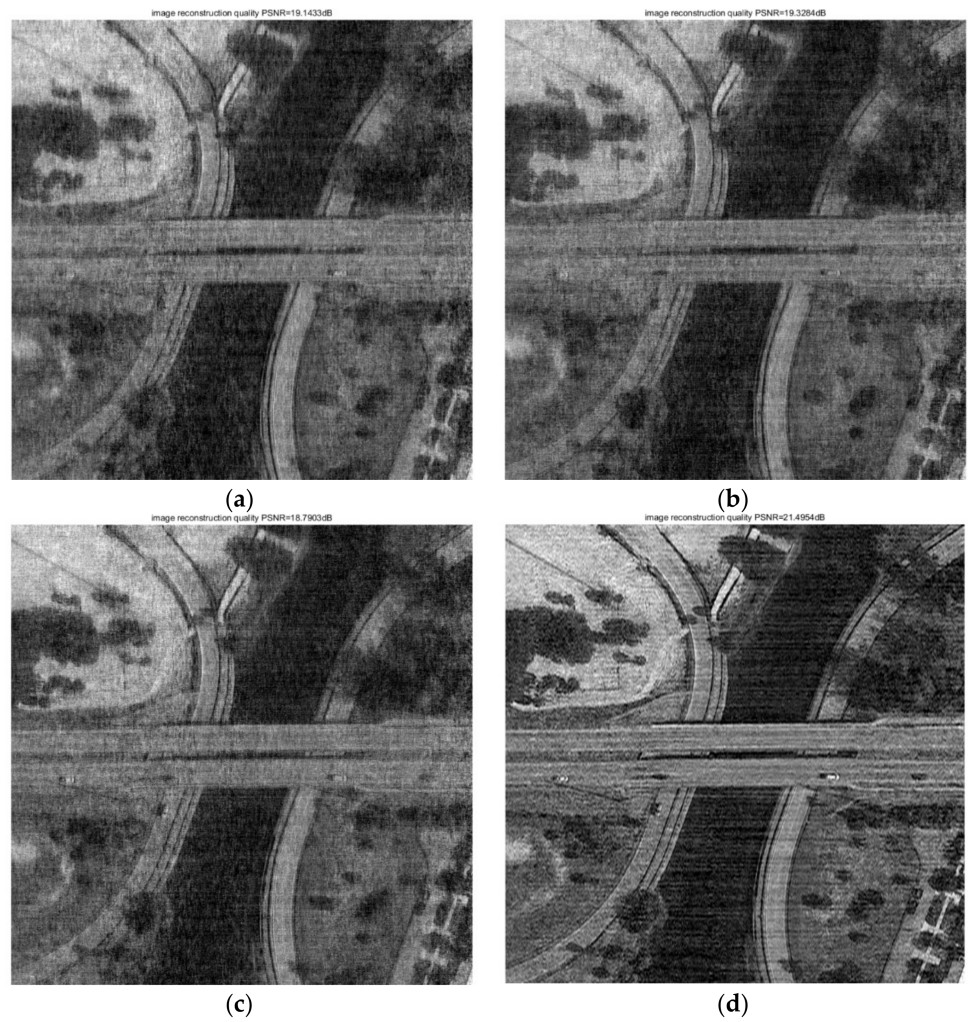

**Figure 4.** Reconstruction results of four different algorithms: (**a**) OMP; (**b**) STOMP; (**c**) GOMP; and (**d**) TL-GOMP.

Table 2 shows simulation results, with the test image at a resolution of $500 \times 500$ pixels selected as the target. The data results show that the $P_{SNR}$ values of the reconstructed images obtained by the TL-GOMP algorithm increased by 18.05% (compared with the GOMP algorithm), 17.82% (compared with the STOMP algorithm), and 18.02% (compared with the OMP algorithm). The $M_{SE}$ values of the images decreased by 53.68%, 53.28%, and 53.62%, respectively.

**Table 2.** $P_{SNR}$, $M_{SE}$, and time of the $500 \times 500$ image reconstructed by OMP series algorithms.

|                   | OMP      | STOMP    | GOMP     | TL-GOMP  |
|-------------------|----------|----------|----------|----------|
| $P_{SNR}$ (dB)    | 18.5193  | 18.5506  | 18.5136  | 21.856   |
| $M_{SE}$          | 914.4386 | 907.8610 | 915.6401 | 424.1134 |
| Running time (s)  | 6.7739   | 3.2574   | 5.5594   | 5.6994   |

Table 3 shows the simulation results, with the test image at a resolution of $650 \times 650$ pixels selected as the target. The data results show that the $P_{SNR}$ values of the reconstructed images obtained by the TL-GOMP algorithm increased by 17.58% (compared with the GOMP algorithm), 15.40% (compared with the STOMP algorithm), and 15.45% (compared with the OMP algorithm). The $M_{SE}$ values of the images decreased by 52.95%, 48.97%, and 49.08%, respectively.

**Table 3.** $P_{SNR}$, $M_{SE}$, and time of the $650 \times 650$ image reconstructed by OMP series algorithms.

|  | OMP | STOMP | GOMP | TL-GOMP |
|---|---|---|---|---|
| $P_{SNR}$ (dB) | 18.9683 | 18.9774 | 18.6246 | 21.8993 |
| $M_{SE}$ | 824.6088 | 822.8859 | 892.5204 | 419.9057 |
| Running time (s) | 12.1603 | 5.5557 | 12.0085 | 11.5920 |

Table 4 shows the simulation results, for which a test image with a resolution of $800 \times 800$ pixels was selected as the target. The resulting data show that the $P_{SNR}$ values of the reconstructed image obtained by the TL-GOMP algorithm increased by 14.40% (compared with the result of the GOMP algorithm), 11.21% (compared with the result of the STOMP algorithm), and 12.29% (compared with the result of the OMP algorithm). The $M_{SE}$ values of the images decreased by 46.36%, 39.28%, and 41.82%, respectively. In the image quality evaluation, the higher the peak signal-to-noise ratio, the better the image quality; in contrast, the smaller the mean square error value, the better the image quality. The red curve shown in Figure 5 is the simulation result of the image reconstruction with different resolutions using the TL-GOMP algorithm. We can observe that image quality improves with an increase in resolution, and thus this algorithm has the advantage of improving the quality of the reconstructed image. Table 5 shows $P_{SNR}$ values and $M_{SE}$ values of the different pixel image repeatedly reconstructed by TL-GOMP algorithms. Table 6 shows $P_{SNR}$ values and $M_{SE}$ values of the different pixel image repeatedly reconstructed by OMP algorithms. Table 7 shows $P_{SNR}$ values and $M_{SE}$ values of the different pixel image repeatedly reconstructed by STOMP algorithms. Table 8 shows $P_{SNR}$ values and $M_{SE}$ values of the different pixel image repeatedly reconstructed by GOMP algorithms. The data results show the advantages of the TL-GOMP image reconstruction algorithm once again.

**Table 4.** $P_{SNR}$, $M_{SE}$, and time of the $800 \times 800$ image reconstructed by OMP series algorithms.

|  | OMP | STOMP | GOMP | TL-GOMP |
|---|---|---|---|---|
| $P_{SNR}$ (dB) | 19.1433 | 19.3284 | 18.7903 | 21.4954 |
| $M_{SE}$ | 792.0467 | 759.0033 | 859.1182 | 460.8311 |
| Running time (s) | 18.8039 | 8.7040 | 22.8444 | 23.6628 |

**Table 5.** $P_{SNR}$, $M_{SE}$, and time of the different pixel image reconstructed by TL-GOMP algorithms.

| $350 \times 350$ Pixel Values | | | $500 \times 500$ Pixel Values | | | $650 \times 650$ Pixel Values | | | $800 \times 800$ Pixel Values | | |
|---|---|---|---|---|---|---|---|---|---|---|---|
| $P_{SNR}$ | $M_{SE}$ | Time | $P_{SNR}$ | $M_{SE}$ | Time | $P_{SNR}$ | $M_{SE}$ | Time | $P_{SNR}$ | $M_{SE}$ | Time |
| 21.3302 | 478.6986 | 3.0450 | 21.9785 | 412.3144 | 5.7769 | 21.7634 | 433.2510 | 12.1419 | 21.4110 | 469.8728 | 24.8091 |
| 21.3607 | 475.3485 | 2.4762 | 21.8199 | 427.6476 | 5.7062 | 21.8976 | 420.0683 | 12.2621 | 21.1822 | 495.2878 | 24.8688 |
| 21.2495 | 487.6785 | 2.4810 | 21.6895 | 440.6868 | 5.7678 | 21.7275 | 436.8474 | 16.2619 | 21.3847 | 472.7259 | 24.6147 |
| 21.3407 | 477.5455 | 2.4567 | 21.9302 | 416.9279 | 5.6955 | 21.9430 | 415.7013 | 12.2062 | 21.2746 | 484.8612 | 24.4856 |
| 21.1693 | 496.7621 | 2.4414 | 21.6719 | 442.4760 | 5.7021 | 21.9532 | 414.7298 | 12.2592 | 21.2543 | 487.1404 | 24.2110 |
| 21.2787 | 484.4030 | 2.4960 | 21.7925 | 430.3612 | 5.8279 | 21.8741 | 422.3505 | 15.7850 | 21.2940 | 482.7085 | 24.5258 |
| 21.2071 | 492.4605 | 2.4652 | 21.8286 | 426.7995 | 5.6639 | 21.7166 | 437.9442 | 15.4912 | 21.3495 | 476.5711 | 24.5294 |
| 21.3974 | 471.3456 | 2.4356 | 21.8625 | 423.4784 | 5.6723 | 21.6169 | 448.1164 | 15.5665 | 21.1921 | 494.1608 | 24.6656 |
| 21.2362 | 489.1702 | 2.4730 | 21.8421 | 425.4695 | 5.6400 | 21.7768 | 431.9201 | 14.7710 | 21.2277 | 490.1291 | 23.7997 |
| 21.1882 | 494.6024 | 2.5031 | 21.8560 | 424.1134 | 5.6994 | 21.8993 | 419.9057 | 11.5920 | 21.4954 | 460.8311 | 23.6628 |
| $P_{SNR}$ **Mean**: 21.2758 | | | $P_{SNR}$ **Mean**: 21.8272 | | | $P_{SNR}$ **Mean**: 21.8168 | | | $P_{SNR}$ **Mean**: 21.3066 | | |
| $M_{SE}$ **Mean**: 484.8015 | | | $M_{SE}$ **Mean**: 427.0275 | | | $M_{SE}$ **Mean**: 428.0835 | | | $M_{SE}$ **Mean**: 481.4289 | | |

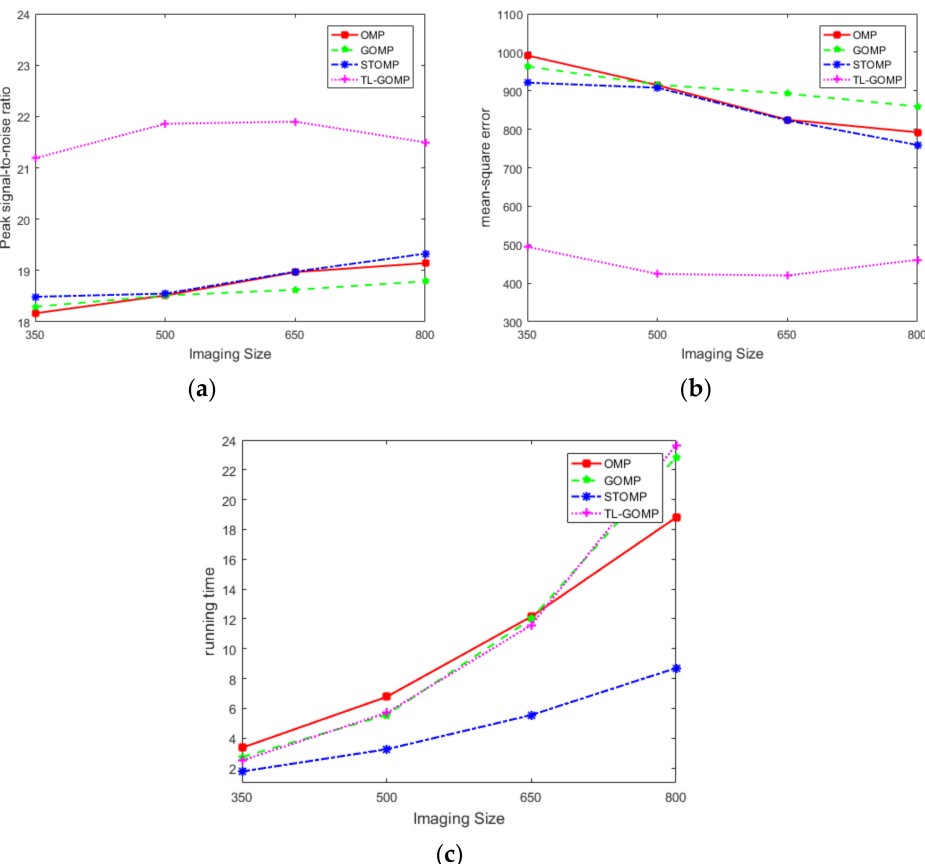

**Figure 5.** (**a**) Relationship between the image size and $P_{SNR}$ of reconstructed images; (**b**) relationship between the image size and $M_{SE}$ of reconstructed images; (**c**) relationship between the image size and running times.

**Table 6.** $P_{SNR}$, $M_{SE}$, and time of the different pixel image reconstructed by OMP algorithms.

| $350 \times 350$ Pixel Values | | | $500 \times 500$ Pixel Values | | | $650 \times 650$ Pixel Values | | | $800 \times 800$ Pixel Values | | |
|---|---|---|---|---|---|---|---|---|---|---|---|
| $P_{SNR}$ | $M_{SE}$ | Time | $P_{SNR}$ | $M_{SE}$ | Time | $P_{SNR}$ | $M_{SE}$ | Time | $P_{SNR}$ | $M_{SE}$ | Time |
| 18.1671 | 991.6670 | 3.9888 | 18.4080 | 938.1606 | 7.5859 | 18.9861 | 821.2361 | 12.3038 | 19.0968 | 800.5658 | 20.3478 |
| 18.2682 | 968.8629 | 3.2198 | 18.5454 | 908.9571 | 7.7925 | 18.9347 | 831.0171 | 12.5650 | 19.1620 | 788.6510 | 21.0002 |
| 18.3101 | 959.5554 | 3.4737 | 18.4466 | 929.8707 | 7.2411 | 18.9399 | 830.0296 | 12.5509 | 19.2571 | 771.5568 | 20.4160 |
| 18.3829 | 943.6068 | 3.6291 | 18.5152 | 915.3030 | 7.1395 | 19.1091 | 798.3158 | 12.4553 | 19.0280 | 813.3486 | 20.9783 |
| 18.0393 | 1021.3 | 3.6883 | 18.4265 | 934.1821 | 7.7281 | 18.9553 | 827.0812 | 12.3023 | 19.2065 | 780.6037 | 22.2521 |
| 18.2515 | 972.5983 | 3.2354 | 18.5735 | 903.0985 | 7.4901 | 18.9398 | 830.0348 | 12.4164 | 19.1368 | 793.2272 | 20.6466 |
| 18.3869 | 942.7279 | 3.2961 | 18.6313 | 891.1402 | 7.5999 | 19.0351 | 812.0306 | 12.4421 | 19.0966 | 800.6090 | 20.4755 |
| 18.2823 | 965.7220 | 3.2445 | 18.6419 | 888.9868 | 6.7575 | 18.8986 | 837.9640 | 12.5608 | 19.1766 | 785.9948 | 20.5140 |
| 18.2712 | 968.1929 | 3.2312 | 18.6113 | 895.2566 | 6.7414 | 19.0885 | 802.0964 | 12.4971 | 19.1135 | 797.4937 | 19.9576 |
| 18.1671 | 991.6670 | 3.3577 | 18.5193 | 914.4386 | 6.7739 | 18.9683 | 824.6088 | 12.1603 | 19.1433 | 792.0467 | 18.8039 |
| $P_{SNR}$ **Mean: 18.2523** | | | $P_{SNR}$ **Mean: 18.5319** | | | $P_{SNR}$ **Mean: 18.9855** | | | $P_{SNR}$ **Mean: 19.1417** | | |
| $M_{SE}$ **Mean: 972.5902** | | | $M_{SE}$ **Mean: 911.9394** | | | $M_{SE}$ **Mean: 821.4414** | | | $M_{SE}$ **Mean: 792.4097** | | |

**Table 7.** $P_{SNR}$, $M_{SE}$, and time of the different pixel image reconstructed by STOMP algorithms.

| $350 \times 350$ Pixel Values | | | $500 \times 500$ Pixel Values | | | $650 \times 650$ Pixel Values | | | $800 \times 800$ Pixel Values | | |
|---|---|---|---|---|---|---|---|---|---|---|---|
| $P_{SNR}$ | $M_{SE}$ | Time | $P_{SNR}$ | $M_{SE}$ | Time | $P_{SNR}$ | $M_{SE}$ | Time | $P_{SNR}$ | $M_{SE}$ | Time |
| 18.4565 | 927.7439 | 1.9871 | 18.8762 | 842.2853 | 3.2992 | 19.1958 | 782.5286 | 5.3546 | 19.4015 | 746.3251 | 8.9414 |
| 18.6146 | 894.5757 | 1.7530 | 18.7574 | 865.6460 | 3.0897 | 19.0214 | 814.5855 | 5.5177 | 19.1294 | 794.5858 | 8.4978 |
| 18.3691 | 946.6053 | 1.7672 | 18.9131 | 835.1674 | 3.2591 | 19.1162 | 797.0074 | 5.3404 | 19.3628 | 753.0119 | 8.3268 |
| 18.7747 | 862.2063 | 1.6509 | 18.6830 | 880.5934 | 3.1428 | 19.0212 | 814.6351 | 5.4728 | 19.3334 | 758.1162 | 8.3945 |
| 18.5523 | 907.5075 | 1.6583 | 18.7680 | 863.5269 | 3.1527 | 19.1821 | 784.9930 | 5.8004 | 19.2737 | 768.6233 | 8.8009 |
| 18.5249 | 913.2449 | 1.7565 | 18.9101 | 835.7386 | 3.1413 | 19.1222 | 795.9122 | 5.2623 | 19.2195 | 778.2766 | 9.1070 |
| 18.4711 | 924.6457 | 1.8162 | 18.9665 | 824.9626 | 3.0769 | 19.1388 | 792.8615 | 5.3666 | 19.1329 | 793.9458 | 8.9904 |
| 18.5027 | 917.9382 | 1.7327 | 18.8543 | 846.5476 | 3.2085 | 19.0365 | 811.7637 | 5.2851 | 19.2444 | 773.8263 | 9.3969 |
| 18.5617 | 905.5508 | 1.6634 | 18.6145 | 894.5940 | 3.1701 | 19.1874 | 784.0488 | 6.3562 | 19.0907 | 801.6980 | 9.0372 |
| 18.4884 | 920.9654 | 1.7663 | 18.5506 | 907.8610 | 3.2574 | 18.9774 | 822.8859 | 5.5557 | 19.3284 | 759.0033 | 8.7040 |
| $P_{SNR}$ **Mean: 18.5316** | | | $P_{SNR}$ **Mean: 18.7894** | | | $P_{SNR}$ **Mean: 19.0999** | | | $P_{SNR}$ **Mean: 19.2517** | | |
| $M_{SE}$ **Mean: 912.0984** | | | $M_{SE}$ **Mean: 859.6923** | | | $M_{SE}$ **Mean: 800.1222** | | | $M_{SE}$ **Mean: 772.7412** | | |

**Table 8.** $P_{SNR}$, $M_{SE}$, and time of the different pixel image reconstructed by GOMP algorithms.

| $350 \times 350$ Pixel Values | | | $500 \times 500$ Pixel Values | | | $650 \times 650$ Pixel Values | | | $800 \times 800$ Pixel Values | | |
|---|---|---|---|---|---|---|---|---|---|---|---|
| $P_{SNR}$ | $M_{SE}$ | Time | $P_{SNR}$ | $M_{SE}$ | Time | $P_{SNR}$ | $M_{SE}$ | Time | $P_{SNR}$ | $M_{SE}$ | Time |
| 18.5342 | 911.3028 | 2.5342 | 18.5828 | 901.1544 | 5.5202 | 18.6002 | 897.5575 | 12.2352 | 18.6504 | 887.2426 | 25.3709 |
| 18.5119 | 915.9863 | 2.3974 | 18.5438 | 909.2880 | 5.2957 | 18.7124 | 874.6603 | 12.7123 | 18.7940 | 858.3842 | 24.2956 |
| 18.4565 | 927.7495 | 2.4619 | 18.5660 | 904.6536 | 5.3261 | 18.7070 | 875.7537 | 11.9305 | 18.6203 | 893.4049 | 25.9419 |
| 18.4604 | 926.9053 | 2.3857 | 18.5033 | 917.7947 | 5.3010 | 18.6821 | 880.7937 | 11.6473 | 18.7787 | 861.4088 | 24.9687 |
| 18.4557 | 927.9302 | 2.4236 | 18.5870 | 900.2903 | 5.2821 | 18.8511 | 847.1601 | 11.7459 | 18.6419 | 888.9819 | 25.1073 |
| 18.5121 | 915.9399 | 2.4229 | 18.4860 | 921.4620 | 5.4175 | 18.6688 | 883.4962 | 11.5865 | 18.7756 | 862.0250 | 25.7629 |
| 18.6042 | 896.7213 | 2.4183 | 18.4826 | 922.1833 | 5.3205 | 18.6093 | 895.6809 | 11.4574 | 18.7594 | 865.2522 | 28.2525 |
| 18.4933 | 919.9185 | 2.4489 | 18.5674 | 904.3666 | 5.2700 | 18.5587 | 906.1624 | 11.5960 | 18.5693 | 903.9659 | 25.3977 |
| 18.3236 | 956.5777 | 2.4210 | 18.5627 | 905.3299 | 5.3125 | 18.6640 | 884.4572 | 11.6323 | 18.6851 | 880.1824 | 24.9104 |
| 18.2944 | 963.0243 | 2.7278 | 18.5136 | 915.6401 | 5.5594 | 18.6246 | 892.5204 | 12.0085 | 18.7903 | 859.1182 | 22.8444 |
| $P_{SNR}$ **Mean: 18.4646** | | | $P_{SNR}$ **Mean: 18.5395** | | | $P_{SNR}$ **Mean: 18.6680** | | | $P_{SNR}$ **Mean: 18.7063** | | |
| $M_{SE}$ **Mean: 926.2056** | | | $M_{SE}$ **Mean: 910.2163** | | | $M_{SE}$ **Mean: 883.8242** | | | $M_{SE}$ **Mean: 875.9966** | | |

### 4.2. Comparison of Simulation Results of the TL-GOMP and Other Algorithms

In this section, to verify the excellent performance of the TL-GOMP algorithm in image reconstruction, we adopt the research method of simulating the same target and comparing the results with those of other algorithms. In the following section, we simulate the test images with resolutions of $350 \times 350$, $500 \times 500$, $650 \times 650$, and $800 \times 800$. Figure 6 shows the simulation results for the test image with a resolution of $800 \times 800$ pixels using different types of algorithms. The figure shows that the image reconstructed by the TL-GOMP algorithm reflects the details or content information of the results. We present the results of the simulation and conduct a quantitative analysis below. In conclusion, the TL-GOMP algorithm has great potential for applications in the field of photonic integrated interference imaging.

In this section, we adopted a test image with a resolution of $350 \times 350$ as the scene target and simulated it using a series of six different image reconstruction algorithms: Compressive Sampling Matching Pursuit (CoSaMP), Generalized Back Propagation (GBP), Iterative Hard Thresholding (IHT), Iteration Reweighted Least Square (IRLS), Subspace Pursuit (SP), and TL-GOMP. The image quality was evaluated using the peak signal-to-noise ratio and mean square error. Table 9 lists the quality evaluation data and code running times of the reconstructed images. From the quantitative data, it can be seen that the $P_{SNR}$ values of the images obtained by the TL-GOMP algorithm are increased by 25.76% (compared with CoSaMP), 10.32% (compared with GBP), 38.50% (compared with IHT), 5.15% (compared with IRLS), and 17.18% (compared with SP). The $M_{SE}$ values of the

images decreased by 63.19%, 36.64%, 74.23%, 21.27%, and 51.09%, respectively, and the code running time was also relatively fast.

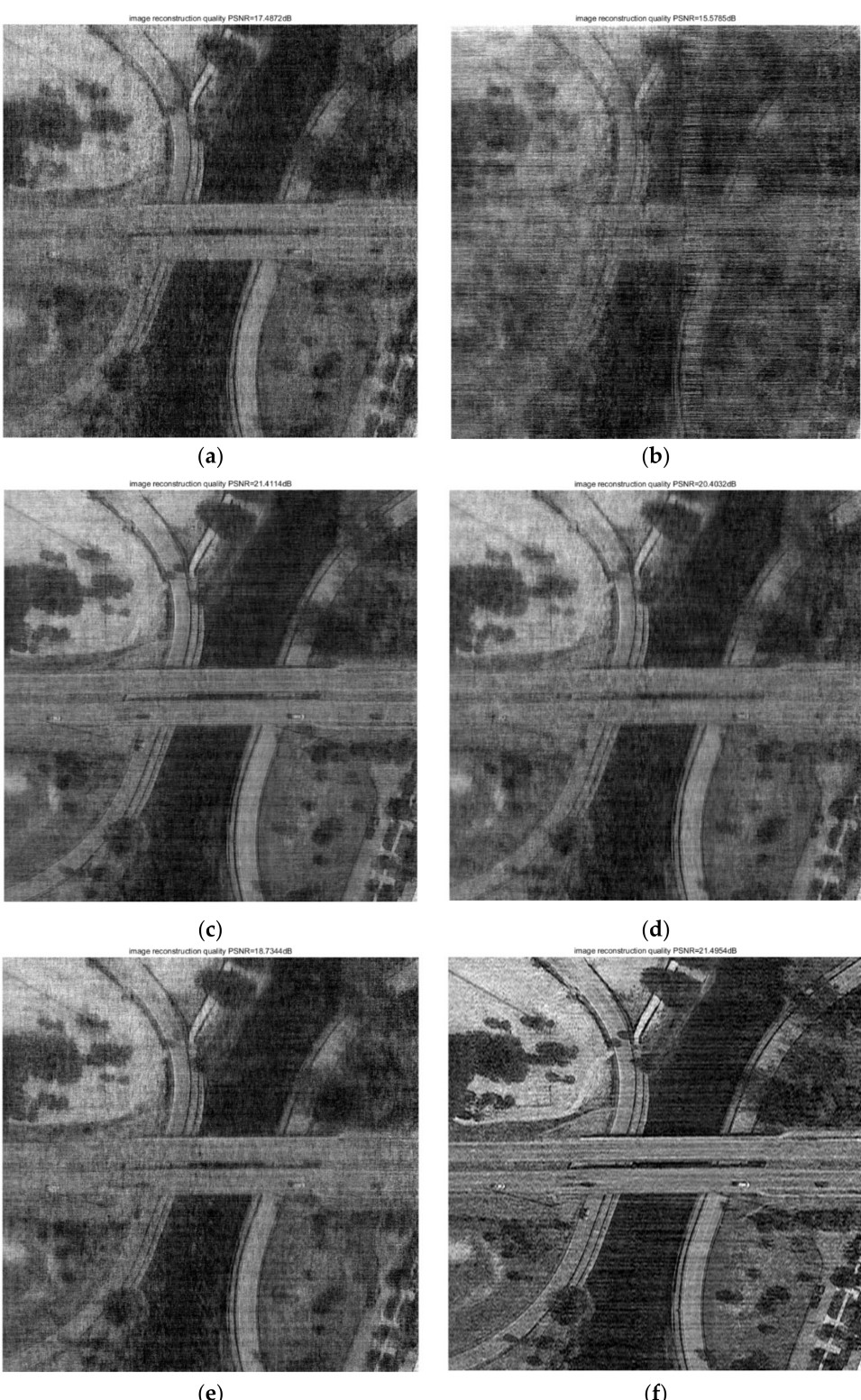

**Figure 6.** Reconstructed image using different algorithms: (**a**) CoSaMP; (**b**) IHT; (**c**) IRLS; (**d**) GBP; (**e**) SP; and (**f**) TL-GOMP.

**Table 9.** $P_{SNR}$, $M_{SE}$, and running time of the 350 × 350 image reconstructed by different algorithms.

|  | CoSaMP | GBP | IHT | IRLS | SP | TL-GOMP |
|---|---|---|---|---|---|---|
| $P_{SNR}$ (dB) | 16.8484 | 19.2067 | 15.2984 | 20.1498 | 18.0819 | 21.1882 |
| $M_{SE}$ | 1343.5 | 780.5690 | 1919.8 | 628.2036 | 1011.3 | 494.6024 |
| Running time (s) | 8.2561 | 15.1944 | 0.9567 | 10.8665 | 6.9127 | 2.5031 |

Table 10 shows the simulation results for the use of a test image with a resolution of 500 × 500 as the target. The results show that the $P_{SNR}$ values of the reconstructed images obtained by the TL-GOMP algorithm improved by 27.74% (compared with CoSaMP), 10.69% (compared with GBP), 42.08% (compared with IHT), 4.01% (compared with IRLS), and 19.75% (compared with SP). The $M_{SE}$ values of the images decreased by 66.47%, 38.51%, 77.47%, 17.64%, and 56.39%, respectively.

**Table 10.** $P_{SNR}$, $M_{SE}$, and running time of the 500 × 500 image reconstructed by different algorithms.

|  | CoSaMP | GBP | IHT | IRLS | SP | TL-GOMP |
|---|---|---|---|---|---|---|
| $P_{SNR}$ (dB) | 17.1099 | 19.7444 | 15.3828 | 21.0134 | 18.2516 | 21.8560 |
| $M_{SE}$ | 1265 | 689.6748 | 1882.8 | 514.9214 | 972.5734 | 424.1134 |
| Running time (s) | 20.4365 | 60.4386 | 2.5600 | 64.1737 | 16.4263 | 5.6994 |

Table 11 shows the simulation results for the use of a test image with a resolution of 650 × 650 as the target. The data in the table show that the $P_{SNR}$ value of the reconstructed image obtained by the TL-GOMP algorithm improved by 26.29% (compared with CoSaMP), 8.87% (compared with GBP), 42.31% (compared with IHT), 2.83% (compared with IRLS), and 17.26% (compared with SP). The $M_{SE}$ values of the images decreased by 64.99%, 33.70%, 77.67%, 12.97%, and 52.39%, respectively.

**Table 11.** $P_{SNR}$, $M_{SE}$, and running time of the 650 × 650 image reconstructed by different algorithms.

|  | CoSaMP | GBP | IHT | IRLS | SP | TL-GOMP |
|---|---|---|---|---|---|---|
| $P_{SNR}$ (dB) | 17.3410 | 20.1147 | 15.3888 | 21.2962 | 18.6761 | 21.8993 |
| $M_{SE}$ | 1199.4 | 633.3042 | 1880.2 | 482.4597 | 881.9959 | 419.9057 |
| Running time (s) | 47.6477 | 151.2679 | 5.5886 | 260.7229 | 34.7940 | 11.5920 |

Table 12 lists the simulation results for the use of the test image with a resolution of 800 × 800 as the target. The data in the table show that the $P_{SNR}$ value of the image reconstructed by the TL-GOMP algorithm increased by 22.92% (compared with CoSaMP), 5.35% (compared with GBP), 37.98% (compared with IHT), 0.40% (compared with IRLS), and 14.74% (compared with SP). The $M_{SE}$ values of the images decreased by 60.26%, 22.23%, 74.40%, 1.91%, and 47.05%, respectively. In Figure 7, the red curve represents the image data as reconstructed by the TL-GOMP algorithm. It can be observed that the peak signal-to-noise ratio and mean square error of the image reconstructed by this algorithm are higher than those reconstructed by the other algorithms. Its mean square error value is much lower than that of the image reconstructed by the other algorithms, and it is also faster in terms of the code running time. In summary, the TL-GOMP algorithm has better image reconstruction performance.

**Table 12.** $P_{SNR}$, $M_{SE}$, and running time of the $800 \times 800$ image reconstructed by different algorithms.

|  | CoSaMP | GBP | IHT | IRLS | SP | TL-GOMP |
|---|---|---|---|---|---|---|
| $P_{SNR}$ (dB) | 17.4872 | 20.4032 | 15.5785 | 21.4114 | 18.7344 | 21.4954 |
| $M_{SE}$ | 1159.7 | 592.5921 | 1799.8 | 469.8253 | 870.2398 | 460.8311 |
| Running time (s) | 96.3305 | 308.2801 | 10.4591 | 650.7766 | 66.1669 | 23.6628 |

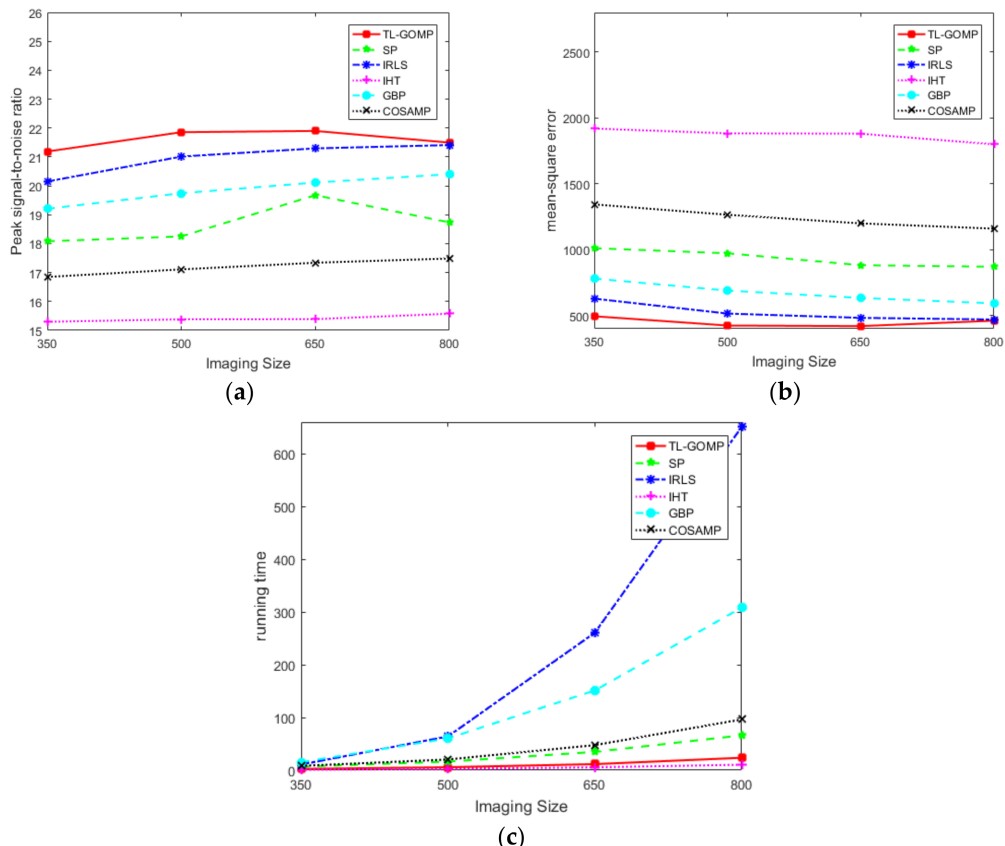

**Figure 7.** (**a**) Relationship between the image size and $P_{SNR}$ of the reconstructed image; (**b**) relationship between the image size and $M_{SE}$ of the reconstructed image; (**c**) relationship between image size and running time.

Table 13 shows $P_{SNR}$ values and $M_{SE}$ values of the different pixel image repeatedly reconstructed by CoSaMP algorithms. Table 14 shows $P_{SNR}$ values and $M_{SE}$ values of the different pixel image repeatedly reconstructed by GBP algorithms. Table 15 shows $P_{SNR}$ values and $M_{SE}$ values of the different pixel image repeatedly reconstructed by IHT algorithms. Table 16 shows $P_{SNR}$ values and $M_{SE}$ values of the different pixel image repeatedly reconstructed by IRLS algorithms. Table 17 shows $P_{SNR}$ values and $M_{SE}$ values of the different pixel image repeatedly reconstructed by SP algorithms. The data results show the advantages of the TL-GOMP image reconstruction algorithm once again.

**Table 13.** $P_{SNR}$, $M_{SE}$, and time of the different pixel image reconstructed by CoSaMP algorithms.

| 350 × 350 Pixel Values | | | 500 × 500 Pixel Values | | | 650 × 650 Pixel Values | | | 800 × 800 Pixel Values | | |
|---|---|---|---|---|---|---|---|---|---|---|---|
| $P_{SNR}$ | $M_{SE}$ | Time | $P_{SNR}$ | $M_{SE}$ | Time | $P_{SNR}$ | $M_{SE}$ | Time | $P_{SNR}$ | $M_{SE}$ | Time |
| 16.8092 | 1355.7 | 8.8949 | 16.9569 | 1310.4 | 21.0424 | 17.4505 | 1169.6 | 53.1953 | 17.4279 | 1175.7 | 104.1132 |
| 16.4940 | 1457.7 | 8.5638 | 17.1272 | 1260 | 20.5154 | 17.3648 | 1192.9 | 51.8239 | 17.5628 | 1139.7 | 100.1575 |
| 16.4971 | 1456.7 | 8.6117 | 17.0410 | 1285.2 | 20.8961 | 17.4114 | 1180.2 | 47.8983 | 17.4618 | 1166.6 | 99.2829 |
| 16.8151 | 1353.8 | 8.6880 | 17.0706 | 1276.5 | 20.2516 | 17.3835 | 1187.8 | 48.2978 | 17.6096 | 1127.5 | 102.0728 |
| 16.7148 | 1385.5 | 8.6742 | 17.1166 | 1263 | 20.0926 | 17.4745 | 1163.1 | 46.9857 | 17.4945 | 1157.8 | 102.7467 |
| 16.5159 | 1450.4 | 8.2333 | 16.9466 | 1313.5 | 20.1809 | 17.3226 | 1204.5 | 48.5044 | 17.6168 | 1125.6 | 106.1725 |
| 16.7907 | 1361.5 | 8.0242 | 17.1552 | 1251.9 | 20.0922 | 17.1436 | 1255.2 | 47.4073 | 17.5932 | 1131.8 | 103.6256 |
| 16.6924 | 1392.7 | 8.1107 | 17.1143 | 1263.7 | 20.1702 | 17.3318 | 1202 | 47.1307 | 17.5843 | 1134.1 | 102.4037 |
| 16.6452 | 1407.9 | 8.0281 | 16.9100 | 1324.6 | 20.1803 | 17.3889 | 1186.3 | 46.7126 | 17.6081 | 1127.9 | 102.3405 |
| 16.8484 | 1343.5 | 8.2561 | 17.1099 | 1265 | 20.4365 | 17.3410 | 1199.4 | 47.6477 | 17.4872 | 1159.7 | 96.3305 |
| $P_{SNR}$ Mean: 16.6823 | | | $P_{SNR}$ Mean: 17.0548 | | | $P_{SNR}$ Mean: 17.3613 | | | $P_{SNR}$ Mean: 17.5446 | | |
| $M_{SE}$ Mean: 1396.54 | | | $M_{SE}$ Mean: 1281.38 | | | $M_{SE}$ Mean: 1194.1 | | | $M_{SE}$ Mean: 1144.64 | | |

**Table 14.** $P_{SNR}$, $M_{SE}$, and time of the different pixel image reconstructed by GBP algorithms.

| 350 × 350 Pixel Values | | | 500 × 500 Pixel Values | | | 650 × 650 Pixel Values | | | 800 × 800 Pixel Values | | |
|---|---|---|---|---|---|---|---|---|---|---|---|
| $P_{SNR}$ | $M_{SE}$ | Time | $P_{SNR}$ | $M_{SE}$ | Time | $P_{SNR}$ | $M_{SE}$ | Time | $P_{SNR}$ | $M_{SE}$ | Time |
| 19.4978 | 729.9666 | 16.7171 | 19.9097 | 663.9063 | 60.8671 | 20.2651 | 611.7477 | 152.6819 | 20.4798 | 582.2355 | 313.1546 |
| 19.1973 | 782.2584 | 15.4194 | 19.8177 | 678.1193 | 58.8448 | 20.2567 | 612.9301 | 151.4318 | 20.4731 | 583.1371 | 307.3742 |
| 19.4500 | 738.0428 | 15.7188 | 19.9173 | 662.7476 | 60.3878 | 20.0630 | 640.8796 | 152.0517 | 20.3483 | 600.1356 | 306.4753 |
| 19.3991 | 746.7401 | 15.4124 | 19.8520 | 672.7897 | 60.0065 | 20.1227 | 632.1369 | 152.8503 | 20.4549 | 585.5920 | 315.1839 |
| 19.3752 | 750.8673 | 15.4889 | 19.7395 | 690.4477 | 59.8597 | 20.2543 | 613.2665 | 151.6609 | 20.4367 | 588.0450 | 307.7829 |
| 19.2693 | 769.3932 | 15.3593 | 19.9330 | 660.3565 | 62.0048 | 20.3432 | 600.8463 | 151.6051 | 20.4022 | 592.7377 | 311.2913 |
| 19.4267 | 742.0074 | 15.5287 | 19.7074 | 695.5632 | 60.5518 | 20.2234 | 617.6454 | 150.1376 | 20.5054 | 578.8216 | 306.9459 |
| 19.3039 | 763.2860 | 15.4373 | 19.7071 | 695.6140 | 60.3061 | 20.2587 | 612.6538 | 150.4258 | 20.3715 | 596.9458 | 308.9359 |
| 19.4260 | 742.1325 | 15.4490 | 19.8390 | 674.8012 | 60.0366 | 20.2990 | 606.9950 | 149.5369 | 20.4816 | 581.9978 | 308.8351 |
| 19.2067 | 780.5690 | 15.1944 | 19.7444 | 689.6748 | 60.4386 | 20.1147 | 633.3042 | 151.2679 | 20.4032 | 592.5921 | 308.2801 |
| $P_{SNR}$ Mean: 19.3552 | | | $P_{SNR}$ Mean: 19.8167 | | | $P_{SNR}$ Mean: 20.2201 | | | $P_{SNR}$ Mean: 20.4357 | | |
| $M_{SE}$ Mean: 754.5263 | | | $M_{SE}$ Mean: 678.4020 | | | $M_{SE}$ Mean: 618.2406 | | | $M_{SE}$ Mean: 588.2240 | | |

**Table 15.** $P_{SNR}$, $M_{SE}$, and time of the different pixel image reconstructed by IHT algorithms.

| 350 × 350 Pixel Values | | | 500 × 500 Pixel Values | | | 650 × 650 Pixel Values | | | 800 × 800 Pixel Values | | |
|---|---|---|---|---|---|---|---|---|---|---|---|
| $P_{SNR}$ | $M_{SE}$ | Time | $P_{SNR}$ | $M_{SE}$ | Time | $P_{SNR}$ | $M_{SE}$ | Time | $P_{SNR}$ | $M_{SE}$ | Time |
| 15.3230 | 1908.9 | 1.0292 | 15.5289 | 1820.5 | 2.7308 | 15.5357 | 1817.6 | 5.8068 | 15.1495 | 1986.7 | 10.6558 |
| 15.2696 | 1932.5 | 0.9657 | 15.3572 | 1893.9 | 2.5608 | 15.5149 | 1826.4 | 5.7885 | 15.6001 | 1790.9 | 10.7756 |
| 15.4349 | 1860.3 | 0.9223 | 15.3438 | 1899.8 | 2.5887 | 15.3108 | 1914.3 | 5.5971 | 15.4884 | 1837.6 | 10.4584 |
| 15.6466 | 1771.8 | 0.9237 | 15.5348 | 1818 | 2.5473 | 15.2720 | 1931.4 | 5.6787 | 15.3543 | 1895.2 | 10.7111 |
| 15.6298 | 1778.7 | 0.9336 | 15.3886 | 1880.3 | 2.5745 | 15.3944 | 1877.8 | 5.5852 | 15.1589 | 1982.4 | 10.4426 |
| 15.4950 | 1834.7 | 0.9343 | 15.6286 | 1779.2 | 2.5784 | 15.4219 | 1865.9 | 5.6056 | 15.2874 | 1924.6 | 10.5495 |
| 15.3671 | 1889.6 | 0.9265 | 15.3710 | 1887.9 | 2.5645 | 15.4774 | 1842.2 | 5.6063 | 15.4666 | 1846.8 | 10.5276 |
| 15.4187 | 1867.3 | 0.9262 | 15.3187 | 1910.8 | 2.5596 | 15.2540 | 1939.5 | 5.6121 | 15.4023 | 1874.4 | 10.4417 |
| 15.6738 | 1760.8 | 0.9315 | 15.6133 | 1785.5 | 2.5755 | 15.3534 | 1895.6 | 5.5838 | 15.4003 | 1875.2 | 10.4518 |
| 15.2984 | 1919.8 | 0.9567 | 15.3828 | 1882.8 | 2.5600 | 15.3888 | 1880.2 | 5.5886 | 15.5785 | 1799.8 | 10.4591 |
| $P_{SNR}$ Mean: 15.4557 | | | $P_{SNR}$ Mean: 15.4468 | | | $P_{SNR}$ Mean: 15.3923 | | | $P_{SNR}$ Mean: 15.3886 | | |
| $M_{SE}$ Mean: 1852.44 | | | $M_{SE}$ Mean: 1855.87 | | | $M_{SE}$ Mean: 1879.09 | | | $M_{SE}$ Mean: 1881.36 | | |

**Table 16.** $P_{SNR}$, $M_{SE}$, and time of the different pixel image reconstructed by IRLS algorithms.

| 350 × 350 Pixel Values | | | 500 × 500 Pixel Values | | | 650 × 650 Pixel Values | | | 800 × 800 Pixel Values | | |
|---|---|---|---|---|---|---|---|---|---|---|---|
| $P_{SNR}$ | $M_{SE}$ | Time | $P_{SNR}$ | $M_{SE}$ | Time | $P_{SNR}$ | $M_{SE}$ | Time | $P_{SNR}$ | $M_{SE}$ | Time |
| 20.6086 | 565.2282 | 11.7302 | 21.2215 | 490.8249 | 65.6006 | 21.3258 | 479.1809 | 270.9207 | 21.3758 | 473.7010 | 664.0105 |
| 20.6022 | 566.0515 | 10.5527 | 21.0093 | 515.4109 | 74.9643 | 21.4131 | 469.6444 | 262.2276 | 21.5207 | 458.1548 | 693.0186 |
| 20.6176 | 564.0528 | 11.4481 | 20.8764 | 531.4202 | 66.3486 | 21.4715 | 463.3728 | 260.0932 | 21.3714 | 474.1812 | 668.7674 |
| 20.4171 | 590.7103 | 10.6213 | 21.0181 | 514.3672 | 64.8940 | 21.6391 | 445.8292 | 261.3721 | 21.5438 | 455.7273 | 662.6378 |
| 20.7530 | 546.7416 | 11.0238 | 21.2248 | 490.4610 | 65.6442 | 21.2211 | 490.8747 | 258.2043 | 21.6174 | 448.0666 | 666.4153 |
| 21.0103 | 515.2937 | 11.2650 | 20.8733 | 531.7981 | 64.7945 | 21.2544 | 487.1234 | 261.8442 | 21.5099 | 459.2951 | 668.0591 |
| 21.1637 | 497.4021 | 11.4127 | 20.9998 | 516.5319 | 64.5561 | 21.2597 | 486.5327 | 255.4833 | 21.3478 | 476.7588 | 677.1501 |
| 20.4757 | 582.7821 | 10.9141 | 21.1686 | 496.8444 | 64.5493 | 21.2293 | 489.9456 | 255.1096 | 21.3538 | 476.1018 | 666.8833 |
| 20.6876 | 555.0313 | 10.6173 | 21.1205 | 502.3765 | 64.5562 | 21.3770 | 473.5656 | 264.4215 | 21.3173 | 480.1209 | 663.1420 |
| 20.1498 | 628.2036 | 10.8665 | 21.0134 | 514.9214 | 64.1737 | 21.2962 | 482.4597 | 260.7229 | 21.4114 | 469.8253 | 650.7766 |
| $P_{SNR}$ Mean: 18.6069 | | | $P_{SNR}$ Mean: 21.0526 | | | $P_{SNR}$ Mean: 21.3487 | | | $P_{SNR}$ Mean: 21.4369 | | |
| $M_{SE}$ Mean: 561.1497 | | | $M_{SE}$ Mean: 510.4957 | | | $M_{SE}$ Mean: 476.8529 | | | $M_{SE}$ Mean: 467.1933 | | |

**Table 17.** $P_{SNR}$, $M_{SE}$, and time of the different pixel image reconstructed by SP algorithms.

| 350 × 350 Pixel Values | | | 500 × 500 Pixel Values | | | 650 × 650 Pixel Values | | | 800 × 800 Pixel Values | | |
|---|---|---|---|---|---|---|---|---|---|---|---|
| $P_{SNR}$ | $M_{SE}$ | Time | $P_{SNR}$ | $M_{SE}$ | Time | $P_{SNR}$ | $M_{SE}$ | Time | $P_{SNR}$ | $M_{SE}$ | Time |
| 17.9046 | 1053.5 | 7.1466 | 18.4409 | 931.0924 | 17.3122 | 18.5753 | 902.7211 | 36.5388 | 18.8447 | 848.4259 | 70.6439 |
| 17.8879 | 1057.5 | 6.9380 | 18.1981 | 984.6217 | 16.6595 | 18.6085 | 895.8334 | 35.2757 | 18.8404 | 849.2576 | 68.9989 |
| 17.8405 | 1069.1 | 7.0076 | 18.1930 | 985.7762 | 16.7420 | 18.5203 | 914.2157 | 36.1913 | 18.8246 | 852.3637 | 74.357 |
| 17.8168 | 1075 | 6.8259 | 18.1758 | 989.6957 | 16.6609 | 18.4910 | 920.4035 | 34.9478 | 18.5796 | 901.8120 | 70.4628 |
| 17.7943 | 1080.6 | 6.7711 | 18.2247 | 978.6191 | 16.6698 | 18.5779 | 902.1786 | 35.1908 | 18.7513 | 866.8650 | 74.8542 |
| 17.7769 | 1084.9 | 6.739 | 18.2543 | 971.9713 | 16.7688 | 18.7444 | 868.2313 | 35.4837 | 18.7457 | 867.9828 | 70.8790 |
| 17.7737 | 1085.7 | 6.6753 | 18.3440 | 952.0871 | 17.0695 | 18.2689 | 968.6950 | 35.2054 | 18.7504 | 867.0365 | 71.0472 |
| 18.0686 | 1014.4 | 6.8211 | 18.1780 | 989.1917 | 17.3466 | 18.7020 | 876.7578 | 35.1299 | 18.7856 | 860.0449 | 69.8509 |
| 17.8676 | 1062.5 | 6.8499 | 18.2797 | 966.3036 | 16.6775 | 18.5849 | 900.7179 | 35.1921 | 18.7113 | 874.8790 | 71.3209 |
| 18.0819 | 1011.3 | 6.9127 | 18.2516 | 972.5734 | 16.4263 | 18.6761 | 881.9959 | 34.7940 | 18.7344 | 870.2398 | 66.1669 |
| $P_{SNR}$ Mean: 17.8813 | | | $P_{SNR}$ Mean: 18.2540 | | | $P_{SNR}$ Mean: 18.5800 | | | $P_{SNR}$ Mean: 18.7568 | | |
| $M_{SE}$ Mean: 1059.45 | | | $M_{SE}$ Mean: 972.1932 | | | $M_{SE}$ Mean: 903.1750 | | | $M_{SE}$ Mean: 865.8907 | | |

*4.3. Simulation Results of Single-Column Signal Reconstruction by the CS TL-GOMP Algorithm*

Figure 8 shows the 256 × 256 target test images. Figure 9 shows the simulation results from the use of the TL-GOMP algorithm to reconstruct a single column of the original signals with different running times. We first selected an image with a resolution of 256 × 256 for testing, arbitrarily selected a 256 × 1 column vector as the original signal, and then performed signal reconstruction 1, 50, 100, and 200 times. The simulation results of the signal reconstruction are shown in Figure 9. It can be observed that the reconstructed signal swings around the original signal and gradually approaches the original signal with an increase in the reconstruction time. Table 18 shows that the residual values of the TL-GOMP algorithm after different runs are 168.5664, 161.6117, 150.3473, and 136.5506 upon comparison of the reconstructed signal with the original signal. Among these, the residual value is an important index for measuring the size of the error or the degree of deviation. The simulation results show that with an increase in the number of original signal reconstructions, the residual value demonstrates a decreasing trend; that is, the accuracy of the reconstructed signal gradually approaches that of the original signal. The TL-GOMP algorithm exhibits good stability in the reconstruction of the original signal. Table 19 shows multiple simulation data with different signal reconstruction times.

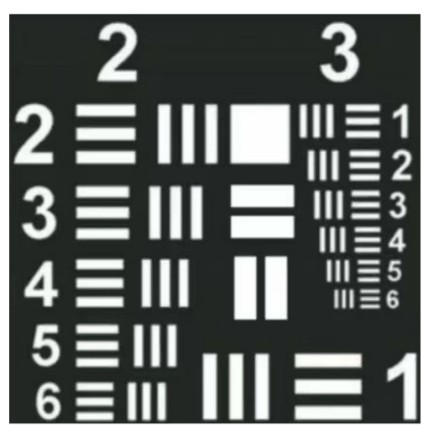

**Figure 8.** Original version of the 256 × 256 image.

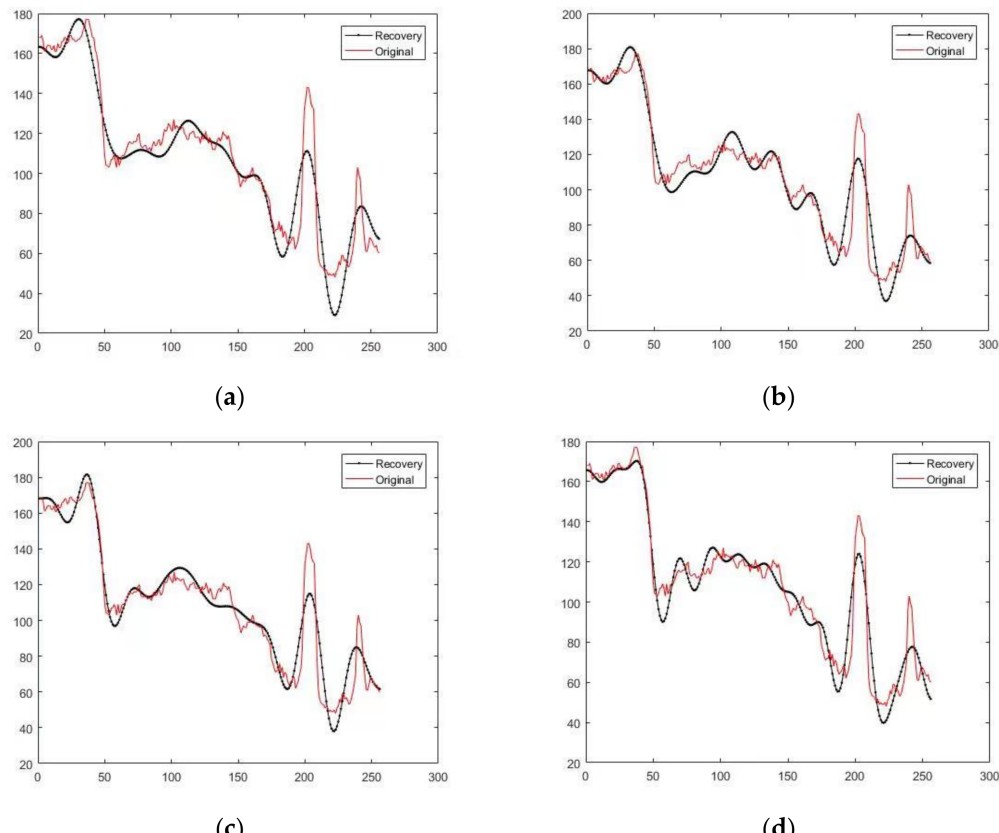

**Figure 9.** Reconstruction results of randomly selected single-column signals: (**a**) 1 time; (**b**) 50 times; (**c**) 100 times; (**d**) 200 times.

**Table 18.** Signal reconstruction times and residual values.

| Times of Signal Reconstruction | 1 | 50 | 100 | 200 |
|---|---|---|---|---|
| Value of residual | 168.5664 | 161.6117 | 150.3473 | 136.5506 |

**Table 19.** Multiple simulation data with different signal reconstruction times.

| | Residual Values | | | | | | | | | |
|---|---|---|---|---|---|---|---|---|---|---|
| 1 time | 168.5664 | 168.6020 | 162.6642 | 165.0314 | 169.6857 | 165.8414 | 158.1089 | 155.4569 | 155.4202 | 149.4011 |
| 50 times | 161.6117 | 159.0616 | 160.1910 | 155.4877 | 155.4719 | 150.6663 | 142.2037 | 149.5014 | 148.7272 | 155.5769 |
| 100 times | 150.3473 | 150.5078 | 150.1486 | 147.8456 | 148.4673 | 153.6050 | 153.2938 | 147.4182 | 155.6388 | 149.4355 |
| 200 times | 136.5506 | 143.0227 | 139.9187 | 143.3128 | 146.7156 | 147.7237 | 146.9315 | 141.7615 | 144.6260 | 143.4051 |

*4.4. Simulation Results of the CS TL-GOMP Algorithm in Image Reconstruction at Different Distances*

We used the Photonic Integrated Circuit to collect the spatial frequency information emitted by the target to form the restoration image. Figure 10 shows the imaging results for the frequency information collected by the microlens array on the PIC at different distances d; the resolution of the restored images is 256 × 256. The "Original image" in Figure 10 represents the restoration image of the microlens array on PIC as the target test image. Because the signal acquisition of the PIC is an under-sampling process, it is necessary to use a sparse signal image reconstruction algorithm in order to recover the content information of the detected target.

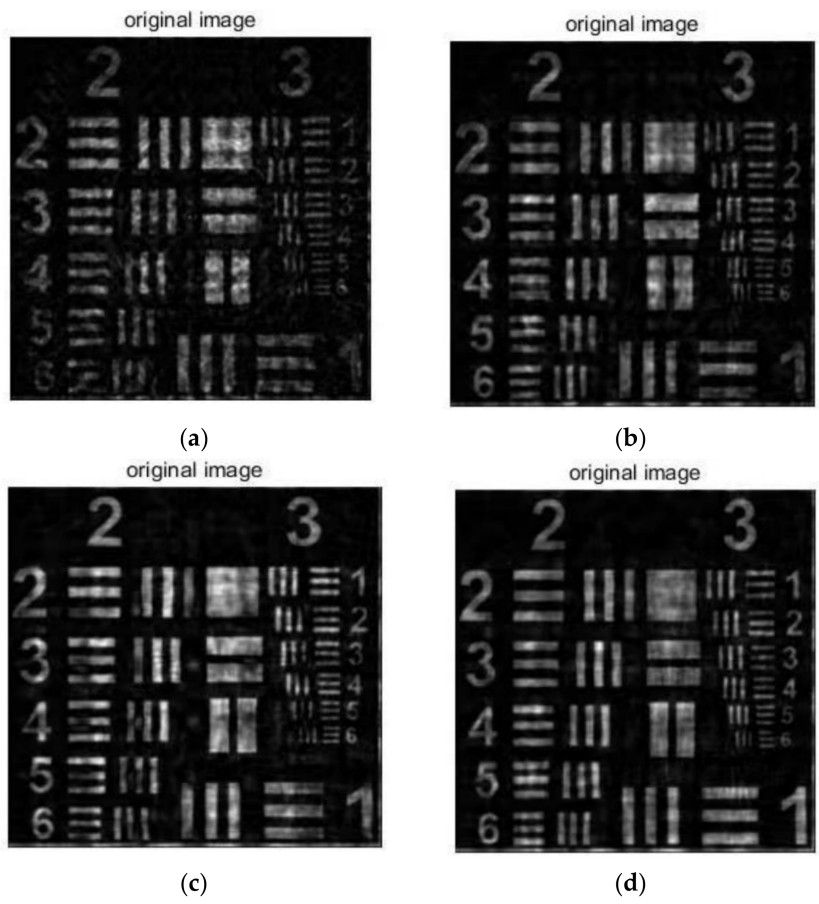

**Figure 10.** Restoration image results of the PIC at different distances: (**a**) d = 75 m; (**b**) d = 125 m; (**c**) d = 175 m; and (**d**) d = 225 m.

In the experiment, the Gaussian random matrix was selected as the measurement matrix, and the discrete cosine transform matrix was used as the sparse matrix. In this part, we conducted several simulation experiments and displayed the experimental data and reconstructed images of one of them, as shown in Figure 11 and Table 20.

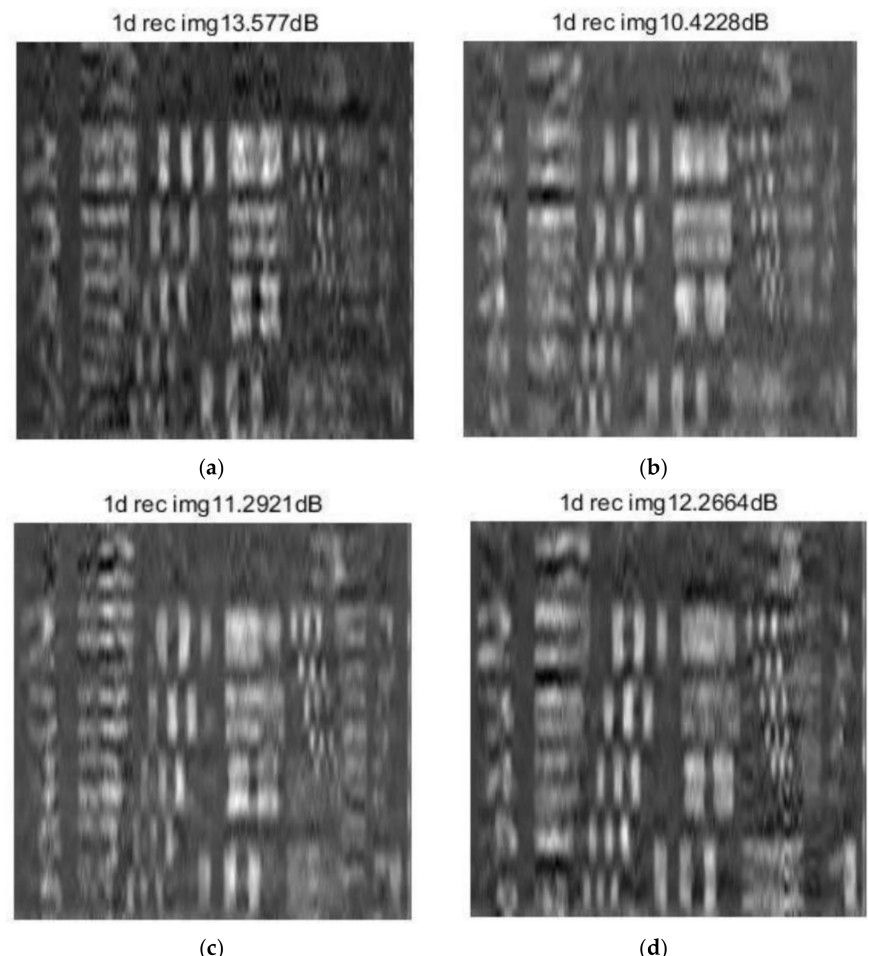

**Figure 11.** Simulation results of the TL-GOMP algorithm reconstruction of restored images at different distances: (**a**) d = 75 m; (**b**) d = 125 m; (**c**) d = 175 m; and (**d**) d = 225 m.

**Table 20.** $P_{SNR}$ and $M_{SE}$ of reconstructed images at different distances.

| d (m) | 75 | 125 | 175 | 225 |
|---|---|---|---|---|
| $P_{SNR}$ (dB) | 13.5770 | 10.4228 | 11.2921 | 12.2664 |
| $M_{SE}$ | $2.8525 \times 10^3$ | $5.8992 \times 10^3$ | $4.8292 \times 10^3$ | $3.8587 \times 10^3$ |

Figure 11 shows the simulation results after the reconstruction of restored images at d = 75, 125, 175, and 225 m using the compressed sensing TL-GOMP algorithm. We used two image quality evaluation indices, the peak signal-to-noise ratio, and the mean square error to measure the image quality. That is, the higher the peak signal-to-noise ratio, the better the image quality. Conversely, the lower the mean square error, the better the image quality. The simulation data in Table 20 show that the compressed sensing TL-GOMP image reconstruction algorithm is suitable for the content restoration of detected targets at different distances. "1d rec img" in Figure 11 represents the result of target reconstruction by TL-GOMP algorithm. We used the image quality evaluation function for evaluation of the reconstructed image; 13.5770 dB, 10.4228 dB, 11.2921 dB, and 12.2664 dB show the peak signal-to-noise ratio of the reconstructed image.

*4.5. Influence of Measurement Number M in the CS TL-GOMP Algorithm*

Figure 8 shows the 256 × 256 target test images. The observation matrix ($M \times N$) $\Phi$ is an important parameter in the CS TL-GOMP algorithm, which can collect the original

signal $\overrightarrow{x}$, obtain the sparse representation $\overrightarrow{\theta}$ by combining with the algorithm, and finally reconstruct the desired signal $\hat{\overrightarrow{\theta}}$. Figure 12a shows the relationship curve of different measurement numbers $M$ on the sparsity $k$ and quality of the reconstructed image. The value of $M$ ranged from 56 to 256, with a step size of 5. The simulation data show that the sparsity $k$ (the number of non-zero values) in the sparse representation $\overrightarrow{\theta}$ is 18. Meanwhile, with an increase in $M$ in the observation vector $\overrightarrow{y}$, the $P_{SNR}$ value of the image also increases. In addition, the $M_{SE}$ value of the image exhibits a decreasing trend. Therefore, we can conclude that the quality of the reconstructed image also improves with an increase in $M$ in the observation vector. In the experiment, the Gaussian random matrix was selected as the measurement matrix, and the discrete cosine transform matrix was used as the sparse matrix; Figure 12 shows the multiple simulation results.

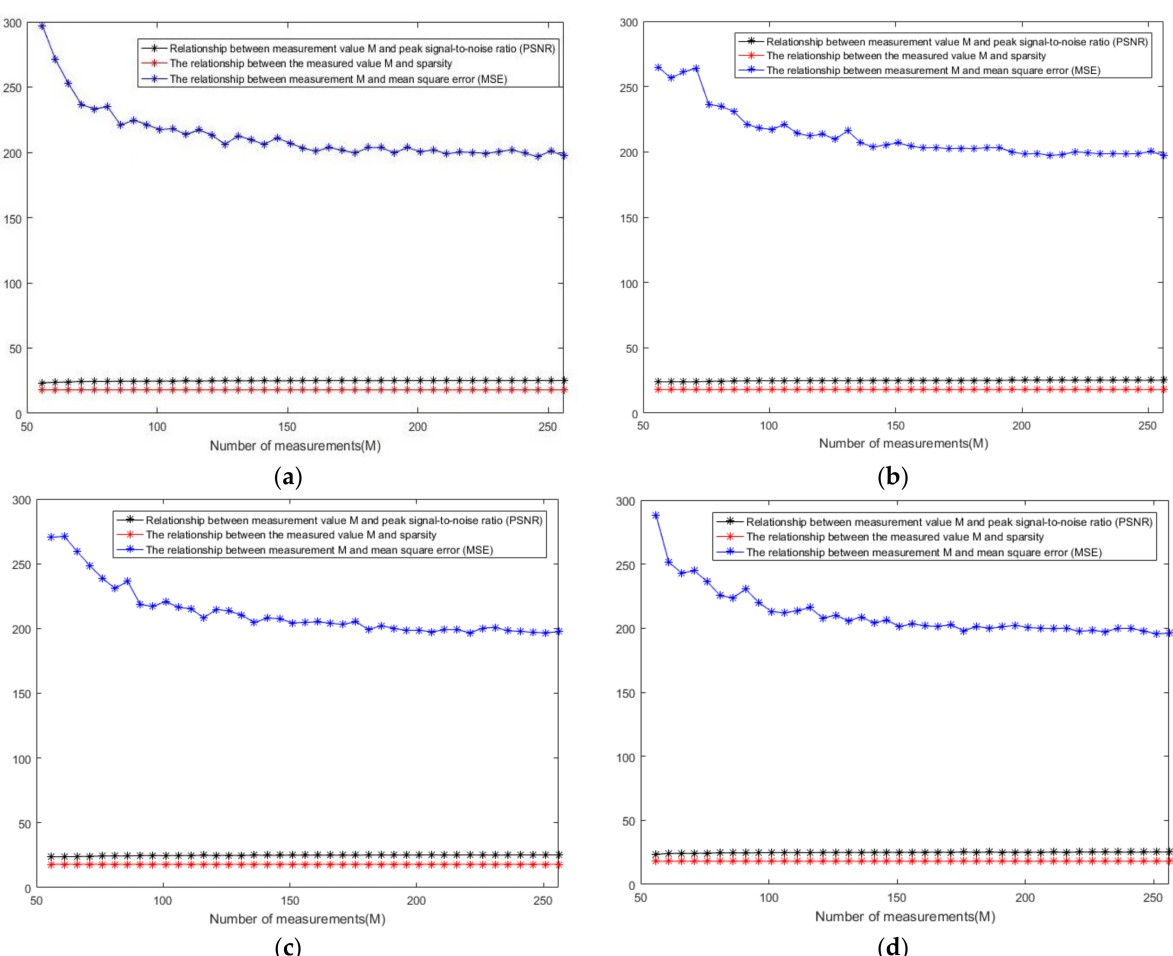

**Figure 12.** Influence of measurement number $M$ on sparsity $k$ and reconstructed image quality. (**a**) The results of the first experiment; (**b**) The results of the second experiment; (**c**) The results of the third experiment; (**d**) The results of the fourth experiment.

*4.6. Influence of Measurement Matrix $M \times N$ and Sparsity $k$ in the CS TL-GOMP Algorithm on the Quality of the Reconstructed Image*

Figure 8 shows the 256 × 256 target test images. Figure 13 shows the results from the simulation of the relationship between different sparsity $k$ values and the quality of the reconstructed image. For the sparsity $k$, we selected the values of 9, 10, 11, 12, and 13 for the simulation of the test image with a resolution of 256 × 256. The results show that the $P_{SNR}$ increases with an increase in $k$, whereas the $M_{SE}$ decreases with an increase in $k$. Figure 14 shows the simulation results for the relationship between the different measurement matrix

sizes and the quality of the reconstructed image. We selected measurement matrices with dimensions of $36 \times 256$, $42 \times 256$, $64 \times 256$, and $85 \times 256$ to simulate the same test image. The simulation results show that $P_{SNR}$ increases with an increase in the size of the size of the measurement matrix. The $M_{SE}$ decreases with an increase in the size of the measurement matrix. Tables 21 and 22 list one of the multiple simulation results. Therefore, we can conclude that the quality of the reconstructed image improves with an increase in *k* and the size of the measurement matrix.

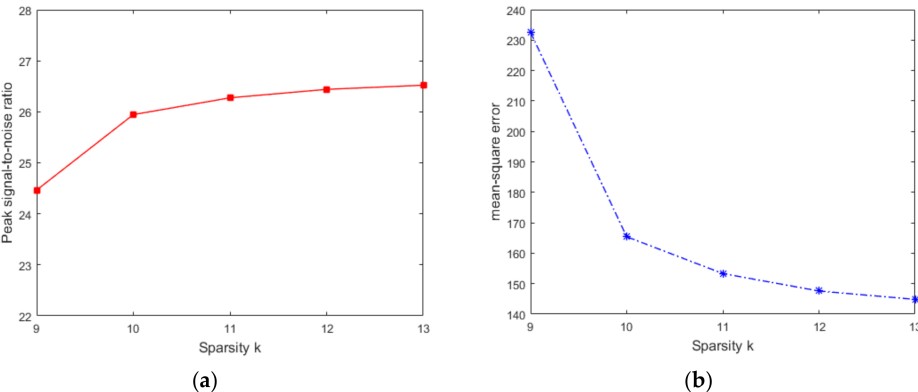

(**a**) (**b**)

**Figure 13.** (**a**) Variation curves for the $P_{SNR}$ of the reconstructed images with sparsity *k*; (**b**) variation curves for the $M_{SE}$ of the reconstructed images with sparsity *k*.

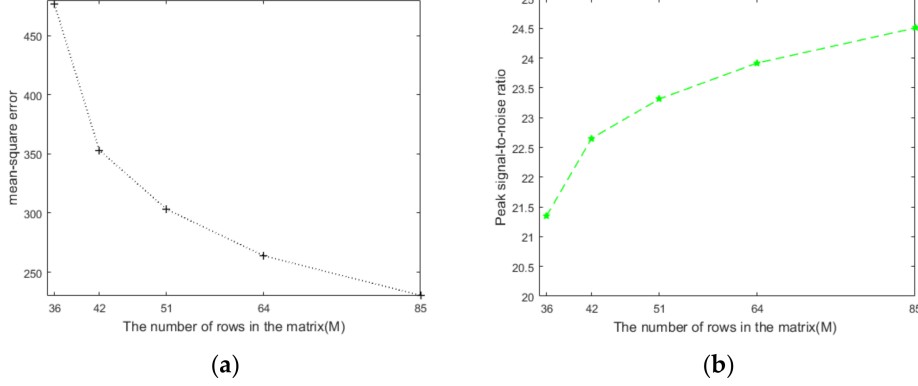

(**a**) (**b**)

**Figure 14.** (**a**) Variation curves for the $M_{SE}$ of reconstructed images with matrix size $M \times N$; (**b**) variation curves for the $P_{SNR}$ of reconstructed images with matrix size $M \times N$.

**Table 21.** $P_{SNR}$ and $M_{SE}$ of the reconstructed images with different sparsity *k*.

| *k* | 9 | 10 | 11 | 12 | 13 |
|---|---|---|---|---|---|
| $P_{SNR}$ | 24.4658 | 25.9451 | 26.2747 | 26.4402 | 26.5216 |
| $M_{SE}$ | 232.5423 | 165.4131 | 153.3254 | 147.5921 | 144.8496 |

**Table 22.** $P_{SNR}$ and $M_{SE}$ of the reconstructed images with different matrix sizes $M \times N$.

| *M* | 85 | 64 | 51 | 42 | 36 |
|---|---|---|---|---|---|
| $P_{SNR}$ | 24.5055 | 23.9166 | 23.3133 | 22.6527 | 21.3493 |
| $M_{SE}$ | 230.4277 | 263.8865 | 303.2150 | 353.0308 | 476.5985 |

We studied the influence of different sparsity and different measurement matrix sizes on reconstruction quality. The measurement matrix selected by us was Gaussian random matrix, and the sparse matrix was discrete cosine transform matrix. For the sparsity *k*,

we selected the values of 9, 10, 11, 12, and 13. In each case of sparsity, we conducted several experiments, and the results of the experiments are shown in Table 23. Similarly, we selected measurement matrices with dimensions of $36 \times 256$, $42 \times 256$, $64 \times 256$, and $85 \times 256$ to simulate the same test image. In the case of the size of each measurement matrix, we also used the same measurement matrix and the same research method to carry out many experiments. The experimental data results are shown in Table 24. According to the data results for many experiments, we conclude that "the quality of the reconstructed image improves with an increase in $k$ and the size of the measurement matrix". However, in order to achieve high-quality image reconstruction, the measurement matrix and sparse matrix of compressed sensing theory are constantly being studied. In future research, we will use an updated measurement matrix to verify the above conclusions.

**Table 23.** $P_{SNR}$ and $M_{SE}$ of the multiple simulation data with different sparsity $k$.

| | | | | | | | | | | |
|---|---|---|---|---|---|---|---|---|---|---|
| k = 9 | $P_{SNR}$ | 24.4658 | 24.4578 | 24.5642 | 24.6083 | 24.5429 | 24.3835 | 24.4639 | 24.4566 | 24.5734 | 24.5788 |
| | $M_{SE}$ | 232.5423 | 232.9702 | 227.3323 | 225.0338 | 228.4496 | 236.9926 | 232.6450 | 233.0364 | 226.8511 | 226.5680 |
| k = 10 | $P_{SNR}$ | 25.9451 | 25.9281 | 25.9644 | 25.9163 | 26.1031 | 26.0670 | 26.1165 | 25.9435 | 26.2481 | 25.9240 |
| | $M_{SE}$ | 165.4131 | 166.0630 | 164.6806 | 166.5131 | 159.5034 | 160.8350 | 159.0111 | 165.4743 | 154.2653 | 166.2188 |
| k = 11 | $P_{SNR}$ | 26.2747 | 26.2508 | 26.3373 | 26.1912 | 26.3492 | 26.2861 | 26.2062 | 26.2744 | 26.2622 | 26.1795 |
| | $M_{SE}$ | 153.3254 | 154.1708 | 151.1292 | 156.3016 | 150.7153 | 152.9223 | 155.7619 | 153.3364 | 153.7671 | 156.7223 |
| k = 12 | $P_{SNR}$ | 26.4402 | 26.4753 | 26.3413 | 26.2574 | 26.5338 | 26.4707 | 26.3457 | 26.4020 | 26.3657 | 26.5155 |
| | $M_{SE}$ | 147.5921 | 146.4039 | 150.9896 | 153.9345 | 144.4438 | 146.5585 | 150.8391 | 148.8964 | 150.1456 | 145.0545 |
| k = 13 | $P_{SNR}$ | 26.5216 | 26.4978 | 26.5798 | 26.6370 | 26.6094 | 26.4142 | 26.6716 | 26.4771 | 26.4774 | 26.6881 |
| | $M_{SE}$ | 144.8496 | 145.6461 | 142.9226 | 141.0527 | 141.9519 | 148.4760 | 139.9344 | 146.3412 | 146.3327 | 139.4024 |

**Table 24.** $P_{SNR}$ and $M_{SE}$ of the multiple simulation data with different matrix size $M \times N$.

| | | | | | | | | | | |
|---|---|---|---|---|---|---|---|---|---|---|
| M = 85 | $P_{SNR}$ | 24.5055 | 24.6425 | 24.5528 | 24.5056 | 24.5522 | 24.5659 | 24.5421 | 24.5434 | 24.4919 | 24.4003 |
| | $M_{SE}$ | 230.4277 | 223.2689 | 227.9270 | 230.4216 | 227.9595 | 227.2443 | 228.4929 | 228.4230 | 231.1488 | 236.0733 |
| M = 64 | $P_{SNR}$ | 23.9166 | 24.1006 | 23.9725 | 24.0803 | 24.1310 | 24.0046 | 24.0576 | 24.1294 | 23.9754 | 24.0435 |
| | $M_{SE}$ | 263.8865 | 252.9398 | 260.5121 | 254.1288 | 251.1774 | 258.5935 | 255.4592 | 251.2689 | 260.3421 | 256.2885 |
| M = 51 | $P_{SNR}$ | 23.3133 | 23.5388 | 23.5970 | 23.4893 | 23.4474 | 23.2988 | 23.2958 | 23.3413 | 23.4279 | 23.3476 |
| | $M_{SE}$ | 303.2150 | 287.8705 | 284.0408 | 291.1752 | 293.9946 | 304.2291 | 304.4373 | 301.2679 | 295.3188 | 300.8316 |
| M = 42 | $P_{SNR}$ | 22.6527 | 22.7587 | 23.1023 | 22.7050 | 22.5485 | 22.8871 | 23.2799 | 22.9354 | 22.9467 | 22.6748 |
| | $M_{SE}$ | 353.0308 | 344.5174 | 318.3073 | 348.8021 | 361.6032 | 334.4826 | 305.5548 | 330.7809 | 329.9219 | 351.2371 |
| M = 36 | $P_{SNR}$ | 21.3493 | 21.9878 | 21.9421 | 21.8499 | 22.1946 | 21.9828 | 21.8089 | 21.8007 | 22.1391 | 21.0789 |
| | $M_{SE}$ | 476.5985 | 411.4304 | 415.7896 | 424.7125 | 392.2991 | 411.9064 | 428.7317 | 429.5465 | 397.3452 | 507.2072 |

## 5. Conclusions

In this study, we improved the traditional image reconstruction algorithm and proposed a TL-GOMP tracing algorithm for compressed sensing. In the simulation, we used the TL-GOMP algorithm and the same series of traditional OMP, STOMP, and GOMP algorithms to perform simulations using the same test targets. The results of the simulation show that the quality of the image reconstructed by the TL-GOMP algorithm was better than that reconstructed by the other traditional algorithms in the same series. To illustrate the advantages of this algorithm more rigorously, we also conducted a comparison simulation between the TL-GOMP algorithm and other image reconstruction algorithms. The results also showed that the quality of the image reconstructed by the TL-GOMP algorithm was better than that reconstructed by the other algorithms, which has potential application value. To verify the stability of the algorithm, we arbitrarily extracted a column

of the original signal column vectors and performed signal reconstruction several times. The simulation results showed that the accuracy of the reconstructed signal gradually approached that of the original signal with an increase in the number of reconstruction runs. The TL-GOMP algorithm was also used to reconstruct the restored images at different detection distances, and the simulation results showed that the algorithm could reproduce the content information of the target. Therefore, the TL-GOMP algorithm is advantageous for applications in photonic integrated interference imaging. It can reconstruct sparse spatial frequency information collected by the PIC and recover the content information of the detected target. In summary, the TL-GOMP algorithm can reconstruct the sparse and unknown information collected as well as recover the content information of unknown targets. This could benefit scientific and technological exploration and production, and it also has good potential for application in the field of photonic integrated interference detection technology.

**Author Contributions:** Conceptualization, C.C., Z.F. and J.Y.; methodology, J.Y.; software, C.C.; validation, Z.W., R.W. and K.L.; formal analysis, S.Y. and B.S.; investigation, J.Y.; resources, C.C.; data curation, J.Y. and Z.F.; writing—original draft preparation, J.Y.; writing—review and editing, Z.F. and C.C.; visualization, C.C.; supervision, Z.W.; project administration, J.Y.; funding acquisition, J.Y. All authors have read and agreed to the published version of the manuscript.

**Funding:** This research received no external funding.

**Data Availability Statement:** The data used to support the findings of this study are available from the corresponding author upon request.

**Acknowledgments:** The authors thank the optical sensing and measurement team of Xidian University for their help. This research was supported by the National Natural Science Foundation of Shaanxi Province (Grant No. 2020JM-206), the National Defense Basic Research Foundation (Grant No. 61428060201), and 111 Project (B17035).

**Conflicts of Interest:** The authors declare no conflict of interest.

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
