# Peer review of "Research on Photon-Integrated Interferometric Remote Sensing Image Reconstruction Based on Compressed Sensing"

_remotesensing, doi:10.3390/rs15092478_

Round 1

Reviewer 1 Report

In this study, an orthogonal matching pursuit algorithm based on compressed sensing theory has been improved for obtaining high-resolution remote sensing images in space exploration. A threshold limited-generalized orthogonal matching pursuit algorithm is proposed. It is applied to reconstruct remote sensing images formed by photon integrated interferometric imaging detectors to improve imaging quality and resolution of detection targets. In this study, TL-GOMP algorithm is used to compare and simulate the same target with OMP algorithm of the same series, and the advantages of this algorithm in image reconstruction are proved. The data results of image quality evaluation function show that the peak signal-to-noise ratio of reconstructed image is increased by 18.02%, and the mean square error is decreased by 53.62%. In the comparison simulation between TL-GOMP algorithm and non-OMP algorithm, TL-GOMP algorithm also shows its effect of image reconstruction.

However, there are still some shortcomings in the article: (1) The pictures in the paper are not clear, as shown in Figure 12. The table in this paper has some format problems. As shown in Table 2, the table form of algorithm flow needs to be adjusted. (2) There are grammatical errors and unsmoothness in the sentences; (3) Please explain the marked information in Figure 10 and Figure 11.

After the authors addressing the above problems, I will suggest to accept this paper titled "Research on Photon-Integrated Interferometric Remote Sensing Image Reconstruction Based on Compressed Sensing".

The Quality of English Language is good.

Author Response

Dear Editors and Reviewers,

Thank you for your letter and for the reviewers’ comments concerning our manuscript entitled “Research on Photon-Integrated Interferometric Remote Sensing Image Reconstruction Based on Compressed Sensing” with the ID remotesensing-2361766. Those comments are all valuable and very helpful for revising and improving our paper, as well as the important guiding significance to our researches. We have studied comments carefully and have made correction which we hope meet with approval. Revised portion are marked in highlight in the paper. The main corrections in the paper and the responds to the reviewer’s comments are as flow:

Responds to the reviewer 1’s comments:

  1. Response to " The pictures in the paper are not clear, as shown in Figure 12. The table in this paper has some format problems. As shown in Table 2, the table form of algorithm flow needs to be adjusted. "

Response: I sincerely thank the reviewers and editors for their valuable comments on my thesis. We have completed improvements to Figure 12 to improve its clarity. At the same time, we also completed the adjustment of the format of the algorithm flow table, as shown in Table 1 and Table 2. Revised portion are marked in highlight in the paper.

  1. Response to " There are grammatical errors and unsmoothness in the sentences. "

Response: I sincerely thank the reviewers and editors for their valuable comments on my thesis. We have carried out further polishing work on the manuscript.

  1. Response to " Please explain the marked information in Figure 10 and Figure 11. "

Response: I sincerely thank the reviewers and editors for their valuable comments on my thesis. We have finished illustrating the marked information in Figures 10 and 11. Revised portion are marked in highlight in the paper.

Reviewer 2 Report

In this manuscript, to achieve high-resolution remote sensing images, a threshold limited-generalized orthogonal matching pursuit (TL-GOMP) algorithm is proposed. With a series of experiments and analyses, the TL-GOMP algorithm can achieve high-quality image reconstruction and has great application potential in photonic integrated interferometric remote sensing detection and imaging. This is an interesting work. To possibly improve the quality of this manuscript, some necessary issues need to be addressed, which are listed as follows.

1.     In section 3, i.e. Methods, excessive descriptions are used to introduce the basic CS theory and OMP algorithm, the authors should simplify this part and highlight the proposed algorithm.

2.     What is the principle of the selection of parameter ms in the TL-GOMP algorithm, and what does the parameter M in qs represent? Please add more details.

3.     The authors should explain more about the advantages of the proposed algorithm in the theory.

4.     In section 4, i.e. Experiments, all experiments should be implemented with multiple Monte Carlo experiments, the authors should state clearly.

5.     To enhance the accuracy of lines, the authors should add more experiments at different imaging sizes.

6.     The conclusion of “the quality of the reconstructed image improves with an increase in k and the size of the measurement matrix.” should be discussed.

7.     Some equations and figures are burdensome, e.g. Figure 1 and Figure 2 seem to be in the same form and can be simplified.

8.     The second component in Figure 2 might be wrong, the sparse matrix and sparse representation should switch the position for each other.

9.     There lacks a complete description of some of the acronyms, e.g. STOMP, GOMP and etc. And some mentioned acronyms are repeatedly explained in the text.

10.  The manuscript needs to be further polished before it can be published on remote sensing, especially some misspellings, tenses, and English usage. For example, “basic pursuit (BP)” must be “basis pursuit (BP)”, the tense of the manuscript should be present tense, the font of some figures are extreme small and etc.

Please check the grammar and spelling carefully.

Author Response

Dear Editors and Reviewers,

Thank you for your letter and for the reviewers’ comments concerning our manuscript entitled “Research on Photon-Integrated Interferometric Remote Sensing Image Reconstruction Based on Compressed Sensing” with the ID remotesensing-2361766. Those comments are all valuable and very helpful for revising and improving our paper, as well as the important guiding significance to our researches. We have studied comments carefully and have made correction which we hope meet with approval. Revised portion are marked in highlight in the paper. The main corrections in the paper and the responds to the reviewer’s comments are as flow:

Responds to the reviewer 2’s comments:

  1. Response to " In section 3, i.e. Methods, excessive descriptions are used to introduce the basic CS theory and OMP algorithm, the authors should simplify this part and highlight the proposed algorithm."

Response: I sincerely thank the reviewers and editors for their valuable comments on my paper. In Section 3 of this paper, we have completed the content simplification of CS theory and OMP algorithm theory.

  1. Response to " What is the principle of the selection of parameter ms in the TL-GOMP algorithm, and what does the parameter M in qs represent? Please add more details. "

Response: I sincerely thank the reviewers and editors for their valuable comments on my paper. We have explained the content information of ms and qs in this paper. the parameter ms can be understood as a variable that controls or adjusts the threshold, which is a range value. The principle of its selection is to constantly change the threshold value and form the corresponding column vector of the first S inner product values in the sensor matrix into a matrix, with the purpose of solving the optimal S least squares solutions to form a sparse representation. After k cycles, the sparsity of the sparse representation is kS. Parameter M represents the number of rows of perception matrix and measurement matrix in compressed sensing theory, or the number of measurements of observation matrix. Revised portion are marked in highlight in section 3.2.

  1. Response to " The authors should explain more about the advantages of the proposed algorithm in the theory. "

Response: I sincerely thank the reviewers and editors for their valuable comments on my paper. We have improved the theory and demonstrated the advantages of the proposed algorithm. The advantage of this algorithm is that the inner product values meeting the threshold conditions can be quickly screened out in time by setting the limiting threshold value, and corresponding column vectors can be directly found in the sensor matrix according to the serial number of the first S inner product values. These inner product values are represented by logical value 1 in the code, while other inner product values are represented by logical value 0. On the other hand, by setting the threshold coefficient ms to adjust the limiting threshold, we constantly combine the serial numbers of the first S inner product values into the corresponding column vectors in the sensor matrix to form a matrix, aiming at solving the optimal S least squares solutions with good universality and flexibility. After k iterations, we can reconstruct the image information of the target through sparse representation with a sparsity of kS. Revised portion are marked in highlight in section 3.2.

  1. Response to " In section 4, i.e. Experiments, all experiments should be implemented with multiple Monte Carlo experiments, the authors should state clearly."

Response: I sincerely thank the reviewers and editors for their valuable comments on my paper. We have improved and demonstrated that the experiment was achieved through multiple Monte Carlo experiments. In all the experiments below, we use the Gaussian random matrix as the measurement matrix, which is established by the randn function in the code, and the values of each element in this matrix satisfy the standard normal distribution. Meanwhile, we use the discrete cosine transform matrix as the sparse matrix, whose function is to sparse representation or compression of the original signal. In the experiment, the measurement matrix will be updated with the operation of the code every time, which reflects the randomness. Therefore, we conducted several simulation experiments in each research part and verified the reliability of the conclusion through the data results. Revised portion are marked in highlight in section 4.

We added more experiments and presented the results of the experimental data in section 4.1. Table 6 shows PSNR values and MSE values of the different pixel image repeatedly reconstructed by TL-GOMP algorithms. Table 7 shows PSNR values and MSE values of the different pixel image repeatedly reconstructed by OMP algorithms. Table 8 shows PSNR values and MSE values of the different pixel image repeatedly reconstructed by STOMP algorithms. Table 9 shows PSNR values and MSE values of the different pixel image repeatedly reconstructed by GOMP algorithms. The data results show the advantages of TL-GOMP image reconstruction algorithm once again. Revised portion are marked in highlight in section 4.1.

We added more experiments and presented the results of the experimental data in section 4.2. Table 15 shows PSNR values and MSE values of the different pixel image repeatedly reconstructed by CoSaMP algorithms. Table 16 shows PSNR values and MSE values of the different pixel image repeatedly reconstructed by GBP algorithms. Table 17 shows PSNR values and MSE values of the different pixel image repeatedly reconstructed by IHT algorithms. Table 18 shows PSNR values and MSE values of the different pixel image repeatedly reconstructed by IRLS algorithms. Table 19 shows PSNR values and MSE values of the different pixel image repeatedly reconstructed by SP algorithms. The data results show the advantages of TL-GOMP image reconstruction algorithm once again. Revised portion are marked in highlight in section 4.2.

We added more experiments and presented the results of the experimental data in section 4.3. Table 21 shows multiple simulation data with different signal reconstruction times. Revised portion are marked in highlight in section 4.3.

In the experiment, the Gaussian random matrix is selected as the measurement matrix, and the discrete cosine transform matrix is used as the sparse matrix. In this part, we conducted several simulation experiments, and displayed the experimental data and reconstructed images of one of them, as shown in Figure 11 and Table 22. Revised portion are marked in highlight in section 4.4.

We added more experiments and presented the results of the experimental data in section 4.5. In the experiment, the Gaussian random matrix is selected as the measurement matrix, and the discrete cosine transform matrix is used as the sparse matrix, Figure 12 shows the multiple simulation results. Revised portion are marked in highlight in section 4.5.

We added more experiments and presented the results of the experimental data in section 4.6. We studied the influence of different sparsity and different measurement matrix size on reconstruction quality. The measurement matrix selected by us is Gaussian random matrix, and the sparse matrix is discrete cosine transform matrix. For the sparsity k, we selected the values of 9, 10, 11, 12, and 13. In each case of sparsity, we conducted several experiments, and the results of experimental data are shown in Table 23. Similarly, we selected measurement matrices with dimensions of 36  256, 42  256, 64  256, and 85  256 to simulate the same test image. in the case of the size of each measurement matrix, we also used the same measurement matrix and the same research method to carry out many experiments. The experimental data results are shown in Table 25. Revised portion are marked in highlight in section 4.6.

  1. Response to " To enhance the accuracy of lines, the authors should add more experiments at different imaging sizes."

Response:I sincerely thank the reviewers and editors for their valuable comments on my paper. In Part 4.1 and 4.2 of this paper, we used images with different resolution sizes respectively to carry out comparative experiments between the same series of OMP image reconstruction algorithms and different image reconstruction algorithms.

  1. Response to " The conclusion of “the quality of the reconstructed image improves with an increase in k and the size of the measurement matrix.” should be discussed. "

Response:I sincerely thank the reviewers and editors for their valuable comments on my thesis. In this part, we studied the influence of different sparsity degree and different measurement matrix size on reconstruction quality. The measurement matrix adopted is Gaussian random matrix, and the research method is Monte Carlo method. We studied the influence of different sparsity degree and different measurement matrix size on reconstruction quality. The measurement matrix selected by us is Gaussian random matrix, and the sparse matrix is discrete cosine transform matrix. In each case of sparsity, we conducted several experiments, and the results of experimental data are shown in Table 23. Similarly, in the case of the size of each measurement matrix, we also used the same measurement matrix and the same research method to carry out many experiments. The experimental data results are shown in Table 25. According to the data results for many times, we conclude that "the quality of reconstructed image is improved with the increase of sparsity and the size of measurement matrix". However, in order to achieve high quality image reconstruction, the measurement matrix and sparse matrix of compressed sensing theory are constantly being studied. In future research, we will use updated measurement matrix to verify the above conclusions. Revised portion are marked in highlight in section 4.6.

  1. Response to " Some equations and figures are burdensome, e.g. Figure 1 and Figure 2 seem to be in the same form and can be simplified."

Response:I sincerely thank the reviewers and editors for their valuable comments on my paper. We have simplified the graphs and equations in the article. Revised portion are marked in highlight in the paper.

  1. Response to " The second component in Figure 2 might be wrong, the sparse matrix and sparse representation should switch the position for each other."

Response:I sincerely thank the reviewers and editors for their valuable comments on my paper. In view of the incorrect representation in the second part of Figure 2, we have converted the sparse matrix and sparse representation into each other's positions. Revised portion are marked in highlight in the paper.

  1. Response to " There lacks a complete description of some of the acronyms, e.g. STOMP, GOMP and etc. And some mentioned acronyms are repeatedly explained in the text."

Response:I sincerely thank the reviewers and editors for their valuable comments on my paper. We have corrected the lack of complete description of some acronyms. Revised portion are marked in highlight in the paper.

  1. Response to " The manuscript needs to be further polished before it can be published on remote sensing, especially some misspellings, tenses, and English usage. For example, “basic pursuit (BP)” must be “basis pursuit (BP)”, the tense of the manuscript should be present tense, the font of some figures are extreme small and etc. "

Response:I sincerely thank the reviewers and editors for their valuable comments on my paper. We have done some further polishing of the manuscript.

Round 2

Reviewer 2 Report

Publish as it is.

Publish as it is.